# MemGen: Weaving Generative Latent Memory for Self-Evolving Agents

**Guibin Zhang, Muxin Fu, Shuicheng Yan**
National University of Singapore

## Abstract

Agent memory shapes how Large Language Model (LLM)-powered agents, akin to the human brain, progressively refine themselves through environment interactions. Existing paradigms remain constrained: parametric memory forcibly adjusts model parameters, and retrieval-based memory externalizes experience into structured databases, yet neither captures the fluid interweaving of reasoning and memory that underlies human cognition. To address this gap, we propose `MemGen`, a dynamic generative memory framework that equips agents with a human-esque cognitive faculty. It consists of a *memory trigger*, which monitors the agent's reasoning state to decide explicit memory invocation, and a *memory weaver*, which takes the agent's current state as stimulus to construct a latent token sequence as machine-native memory to enrich its reasoning. In this way, `MemGen` enables agents to recall and augment latent memory throughout reasoning, producing a tightly interwoven cycle of memory and cognition. Extensive experiments across eight benchmarks show that `MemGen` surpasses leading external memory systems such as ExpeL and AWM by up to $38.22\%$, exceeds GRPO by up to $13.44\%$, and exhibits strong cross-domain generalization ability. More importantly, we find that without explicit supervision, `MemGen` spontaneously evolves distinct human-like memory faculties, including planning memory, procedural memory, and working memory, suggesting an emergent trajectory toward more naturalistic forms of machine cognition. Codes are available at https://github.com/bingreeky/MemGen.

## 1 Introduction

The ascent of Large Language Model (LLM)-powered agents marks a paradigm shift across diverse domains (Luo et al., 2025b; Yang et al., 2024b; Qian et al., 2025; Singh et al., 2025; Pantiukhin et al., 2025; Ren et al., 2025). Pivotal to this success is the concept of *agent memory* (Zhang et al., 2024b; Wu et al., 2025b), which enables LLM agents to learn progressively from environmental interactions (Zhang et al., 2025a; Qiu et al., 2025b). Crucially, this conception of agent memory extends beyond that of conversational agents (*i.e.*, personalized memory (Wu et al., 2025b)), whose primary role is to sustain coherence across long-horizon, multi-turn dialogues (Chhikara et al., 2025; Xu et al., 2025a; Packer et al., 2024; Zhong et al., 2023). Rather, the scope of this paper is primarily on enabling agents to internalize experience, simulate human-like cognitive iteration, and progressively enhance problem-solving competence (Gao et al., 2025).

The memory serving as this self-evolving engine typically manifests in two dominant paradigms. The first is **(I) parametric memory**, which internalizes experiences by directly updating agents' parameters (Yao et al., 2024; Yang et al., 2023; Zeng et al., 2023; Chen et al., 2024b; 2025). While this approach can yield substantial performance gains, its reliance on parameter modification inevitably entails *catastrophic forgetting*, *i.e.*, the erosion of general knowledge (Dou et al., 2024). Conversely, the second paradigm is **(II) retrieval-based memory**, which externalizes past experiences into a structured database, such as *(i) raw trajectories* (Luo et al., 2025a; Zhang et al., 2025a; Zhao et al., 2024), *(ii) high-level experiences* (Zhao et al., 2024; Tang et al., 2025; Fang et al., 2025; Wang et al., 2024c), and *(iii) condensed skills* like reusable APIs (Zheng et al., 2025) or MCP boxes (Qiu et al., 2025b;a). Although this non-invasive approach circumvents catastrophic forgetting, its efficacy is fundamentally tethered to context engineering. It adheres to a rigid execution pipeline, providing retrieved context to the agent without achieving the fluid, seamless integration characteristic of truly internalized memory (Su et al., 2025).

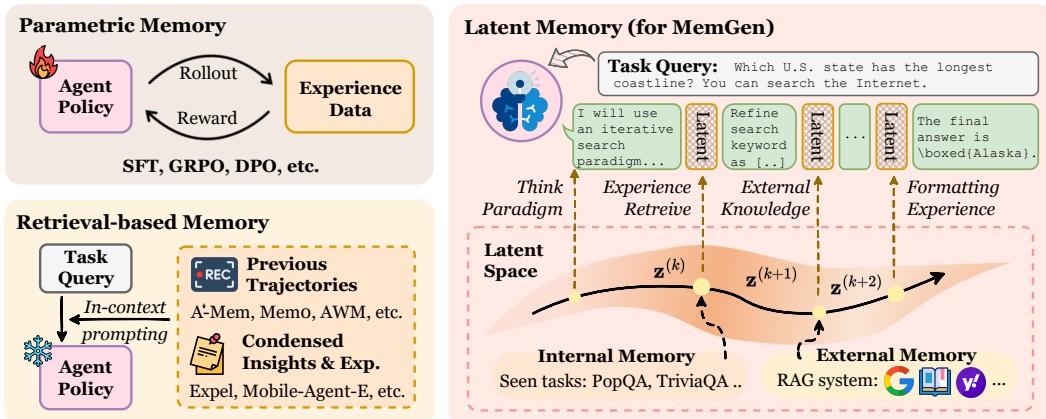

Figure 1: The comparison among parametric memory, retrieval-based memory and `MemGen`. We drew inspiration from the layout presented in Figure 1 of Li et al. (2025a).

Given these deficiencies, **latent memory** offers a compelling alternative, leveraging *latent states* as a machine-native, high-density medium for memory. Existing approaches either use the **(i) key-value (KV) cache** to maintain dynamic memory set (Gim et al., 2024; Jin et al., 2024; Hongkang Yang et al., 2024), yet which is primarily confined to addressing long-context issues, or **(ii) latent token embeddings** to store agent experiences (Wang et al., 2024b; 2025b), which still rely on invasive LLM parameter updates. More fundamentally, these methods diverge from human cognition in two critical dimensions: they lack the seamless interleaving of reasoning and memory, a process where thought and memory dynamically reshape one another, and remain largely retrieval-based, fetching memories by embedding similarity (Wang et al., 2024b) rather than generatively reconstructing them into novel, coherent insights. This leads to the pivotal research question that motivates our work:

> *How can we architect agent memory as a dynamic cognitive faculty, capable of fluid, reconstructive processes that interweave seamlessly with reasoning?*

To address this challenge, we introduce `MemGen`, a dynamic and generative memory framework designed to endow any LLM agent with a more human-esque cognitive faculty. At its core, `MemGen` continuously monitors an agent's cognitive state, enabling it to dynamically invoke a generative process that synthesizes a bespoke latent memory at any critical juncture during its reasoning process. Practically, `MemGen` comprises two synergistic components: a reinforcement learning (RL)-trained ♣ **memory trigger**, which acts as a metacognitive monitor to discern the opportune moments for explicit memory invocation; and a ♠ **memory weaver**, which takes the agent's current state as a stimulus to draw upon relevant implicit parametric memory (potentially augmented with externally retrieved information) and then reconstructs this synthesis into a succinct, machine-native *latent memory*. With the reasoning core fixed, `MemGen` inherently mitigates catastrophic forgetting when exposed to new data, and equips agents with a generative memory deeply integrated with reasoning.

**Experimental Observation.** Extensive experiments across nine benchmarks and four baseline categories demonstrate that `MemGen` delivers ❶ **substantial performance gains**, with improvements of up to $31.7\%$ on ALFWorld (Shridhar et al., 2021) and $27.1\%$ on KodCode (Xu et al., 2025d) with `Qwen3-8B`, surpassing parametric memory (REINFORCE++, $+5.8\%$) and the GRPO method ($+5.32\%$); ❷ **strong cross-domain generalization**, where training in the math domain not only avoids degradation elsewhere but also boosts performance in science reasoning ($+6.06\%$) and code generation ($+5.1\%$); and ❸ **continual learning ability**, maintaining stable performance in previously trained domains even after fine-tuning on three additional ones.

**Analysis & Interpretation.** Beyond quantitative evaluation, we also analyze the functional behavior of the latent memories learned by `MemGen`. Through post-hoc interventions examining the impact of removing specific latent memory on different agent failure modes, we found that different latent memory tokens play distinct computational roles within the agent's reasoning process, including ❶ **planning memory**, where certain latent tokens specifically support high-level task planning, ❷ **procedural memory**, where some latent memory tokens facilitate the agent's recall of task-specific procedural skills, such as tool usage and answer formatting, and ❸ **working memory**, where certain tokens help the agent maintain coherence and understanding over long contexts

within a single task session. These specializations strongly reveal that `MemGen` endows the agent with precise, functionally distinct memory.

## 2 RELATED WORK

**LLM & Agent Memory.** As outlined in §1, existing memory mechanisms designed to evolve the problem-solving capacity of LLM agents can be broadly categorized into three classes: **(I) parametric memory**, which either integrates past experiences directly into agent parameters through finetuning, as in FireAct (Chen et al., 2023), AgentLumos (Yin et al., 2024), and others (Zhang et al., 2024a; Fu et al., 2025), or maintains them in external parameter modules (Tack et al., 2024; Wang et al., 2024a); **(II) retrieval-based memory**, which abstracts prior experiences into transferable knowledge (Zhang et al., 2025a; Zhao et al., 2024), or distills them into reusable tools and skills (Zheng et al., 2025; Wang et al., 2025c; Qiu et al., 2025b;a); and **(III) latent memory**, which leverages implicit representations to encode and retrieve experience (Wang et al., 2024b; 2025b; Hu et al., 2025b; Liu et al., 2024; Sun et al., 2025). Our `MemGen` falls within the latent memory paradigm, yet distinguishes itself from prior approaches through its interweaving of reasoning and memory, as well as generative reconstruction mechanism.

**Latent Computation.** Our method is also closely related to latent computation, wherein latent states are employed to intervene in or reshape the LLM's reasoning process (Zhu et al., 2025). Prominent paradigms include: **(I) architecturally enabling native latent reasoning**, exemplified by Coconut (Hao et al., 2024), CODI (Shen et al., 2025b), LatentR3 (Zhang et al., 2025b) and CoLaR (Tan et al., 2025), which render the LLM's inference process inherently latent and machine-native; and **(II) employing latent computation to steer LLM generation**, as in LaRS (Xu et al., 2023), LatentSeek (Li et al., 2025a), SoftCoT (Xu et al., 2025c;b), and Coprocessor (Liu et al., 2024), which leverage latent representations to modulate the quality of generated outputs.

## 3 PRELIMINARY

**Notation.** We formalize the agent's interaction within an environment $\mathcal{E}$. An agent, powered by an LLM parameterized by $\theta$, is denoted as $\pi_\theta$. For a given task $\mathbf{x}$, the agent's interaction unfolds as a high-level trajectory, denoted as follows $\tau = (s_0, a_0, s_1, a_1, \ldots, s_T)$, where $s_t$ represents the state of the environment and $a_t$ is the high-level action taken by the agent. More internally, each action $a_t$ is essentially a sequence of tokens, $a_t = (\mathbf{z}_{t,1}, \mathbf{z}_{t,2}, \ldots, \mathbf{z}_{t,L_t})$, generated autoregressively by the LLM. The generation of the $j$-th token is conditioned on the current state $s_t$ and all previously generated tokens within that action:

$$\mathbf{z}_{t,j} \sim \pi_\theta(\cdot \mid s_t, \mathbf{z}_{t,<j}). \tag{1}$$

After an entire action sequence $\mathbf{a}_t$ is generated, it is executed in the environment, which transitions the state from $s_t$ to $s_{t+1}$. The success of the trajectory $\tau$ is evaluated by a reward function $R(\tau)$.

**Problem Formalization** Given a history of past experiences $\mathcal{H} = \{(x_i, \tau_i)\}_{i=1}^N$, the objective is to leverage this history to maximize the agent's performance on new tasks. The policy $\pi_\theta$ and a memory system $\mathcal{M}$ are thus jointly optimized to maximize the expected reward over a task distribution $\mathcal{D}$:

$$\max_{\theta, \mathcal{M}} \ \mathbb{E}_{x \sim \mathcal{D}, \ \tau \sim \pi_\theta, \mathcal{M}} \big[ R(\tau) \big], \tag{2}$$

during which $\mathcal{M}$ is to produce a memory representation, $m$, which conditions the agent's policy. The action at any timestep $t$ is thus sampled as $a_t \sim \pi_\theta(\cdot \mid s_t, m_t)$, where $m_t$ is the inserted memory at that step. Crucially, the nature and timing of memory generation, which we denote as the function $f_\mathcal{M}$, vary across different paradigms. We express the generation of the memory $m_t$ as:

$$m_t = f_\mathcal{M}(s_t, \mathcal{H}, m_{<t}), \tag{3}$$

which accommodates diverse memory invocation granularities. For task-level memory (*e.g.*, Expel (Zhao et al., 2024) and G-Memory (Zhang et al., 2025a)), $f_\mathcal{M}$ is invoked only at $t = 0$, and $m_t = m_0$ for all subsequent steps. For step-level memory (*e.g.*, AgentKB (Tang et al., 2025)), $f_\mathcal{M}$ is invoked at every step $t$ to update the memory. In parametric memory, the influence of $\mathcal{H}$ is compiled into $\theta$, rendering memory generation implicit in the model parameters. Our work, which introduces dynamic latent memory, focuses on designing a more fine-grained $f_\mathcal{M}$ that decides for itself the optimal moments to regenerate $m_t$ at the token level during the agent's reasoning process.

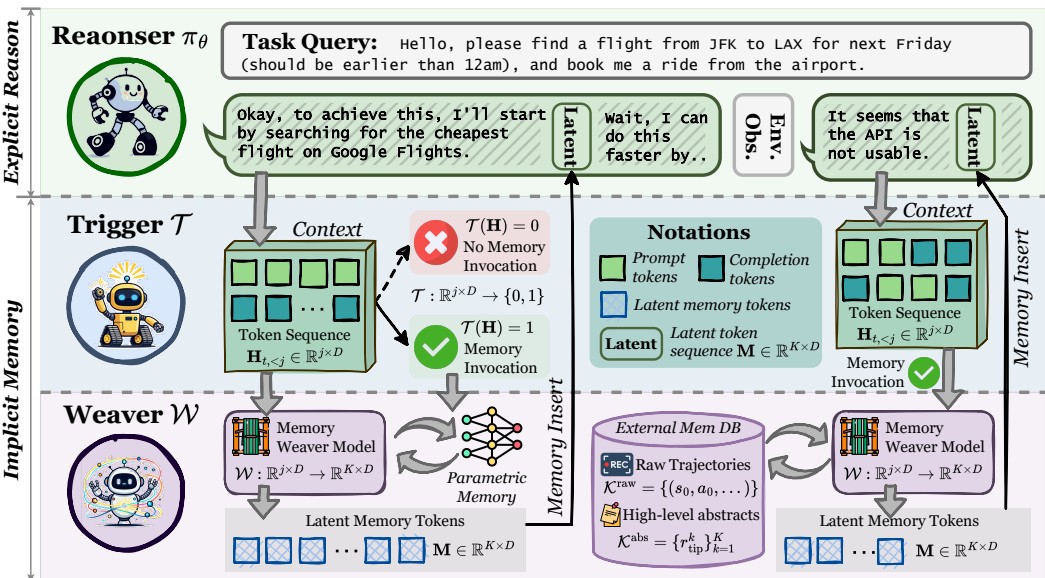

Figure 2: The overview of our proposed `MemGen`.

## 4 METHODOLOGY

### 4.1 `MemGen`: INTERLEAVING MEMORY AND REASONING

Just as *a person is the sum of their past experiences* (McAdams, 2001), memory profoundly shapes an agent's behavior (Xiong et al., 2025). Yet existing agent memory systems lack the adaptive fluidity characteristic of human cognition. Human reasoning and recollection operate in a continuous interplay, whereas most agent memory frameworks retrieve information only once at task initiation and append it to the query in a coarse and static manner. `MemGen` is designed to close this gap by enabling memory to participate dynamically in the reasoning process.

As shown in Figure 2, the reasoning process in an agent equipped with `MemGen` unfolds autoregressively, driven by a frozen core LLM, the reasoner $\pi_\theta$. For a given state $s_t$, $\pi_\theta$ generates the action $a_t = (\mathbf{z}_{t,1}, \ldots, \mathbf{z}_{t,L_t})$. `MemGen` continuously monitors the token-by-token generation process and performs on-demand memory insertion. At each token-generation step $j$, a **memory trigger** $\mathcal{T}_{\text{trigger}}$ monitors the reasoner's internal cognitive state to determine if a moment of reflection, *i.e.*, a memory invocation, is necessary. Specifically, as the reasoner generates the token sequence $\mathbf{z}_{t,<j}$, it produces a corresponding sequence of hidden state vectors, $\mathbf{H}_{t,<j} = (\mathbf{h}_{t,1}, \ldots, \mathbf{h}_{t,j-1})$, where each $\mathbf{h}_{t,k} \in \mathbb{R}^{d_{\text{model}}}$. The trigger takes the current hidden states $\mathbf{H}_{t,<j}$ as a representation of the reasoner's current metacognitive state and computes an invocation probability:

$$p_j = \sigma\left(\mathcal{T}_{\text{trigger}}(\mathbf{h}_{t,1}, \ldots, \mathbf{h}_{t,j-1})\right), \tag{4}$$

from which a binary decision, $d_j \sim \text{Bernoulli}(p_j) \in \{\texttt{INVOKE}, \texttt{SKIP}\}$, is sampled. If the decision is to [SKIP], $\pi_\theta$ proceeds with its standard autoregressive generation, *i.e.*, $\mathbf{z}_{t,j} \sim \pi_\theta(\cdot \mid s_t, \mathbf{z}_{t,<j})$. However, if the decision is to $\texttt{INVOKE}$, the reasoning process is momentarily paused. This summons the second core component of our framework: the **memory weaver** $\mathcal{W}_{\text{weaver}}$, which takes the same cognitive state $\mathbf{H}_{t,<j}$ as a stimulus to perform a generative act of recollection. It synthesizes a bespoke, machine-native *latent memory*, formalized as $\mathbf{M}_t \in \mathbb{R}^{K \times d_{\text{model}}}$ with fixed length $K$:

$$\mathbf{M}_t := [\mathbf{m}_{t,1}, \mathbf{m}_{t,2}, \cdots, \mathbf{m}_{t,K}] = \mathcal{W}_{\text{weaver}}(\mathbf{H}_{t,<j}), \tag{5}$$

where the memory is generated not merely from the parametric knowledge encoded within $\mathcal{W}_{\text{weaver}}$ but may also incorporate cues retrieved from external memory databases (detailed implementation is elaborated in §4.3). Crucially, $\mathbf{M}_t$ is not a verbatim restatement of prior content but a selective reconstruction, filtered and integrated through $\mathcal{W}_{\text{weaver}}$, akin to the memory consolidation process in human brain (Spens & Burgess, 2024). Once formed, the latent memory is woven seamlessly into the reasoner's ongoing dynamics: its hidden states are prepended to $\mathbf{H}_{t,<j}$, upon which the reasoner resumes generation conditioned on this enriched context,

$$\mathbf{z}_{t,j} \sim \pi_\theta(\cdot \mid s_t, \mathbf{z}_{t,<j}, \mathbf{M}_t). \tag{6}$$

This iterative cycle of generation, monitoring, invocation, weaving, and reintegration elevates reasoning from a linear unfolding to a recursive dialogue with memory, all without altering the frozen reasoner $\pi_\theta$, and thereby preserving its general capabilities. In the following sections, we detail the implementations of the memory trigger ($\triangleright$ §4.2) and the memory weaver ($\triangleright$ §4.3).

## 4.2   MEMORY TRIGGER: LEARNING TO INVOCATE MEMORY

In this section, we describe the concrete instantiation and training recipe of the memory trigger $\mathcal{T}_{\text{trigger}}$. Recall from §4.1 that $\mathcal{T}_{\text{trigger}}$ serves as a metacognitive monitor, observing the evolving reasoning state of the frozen reasoner $\pi_\theta$ and producing a binary decision $d_j \in \{\texttt{INVOKE}, \texttt{SKIP}\}(0 \leftrightarrow \texttt{SKIP}, 1 \leftrightarrow \texttt{INVOKE})$ that determines whether the memory weaver should be invoked at token $j$.

**Instantiation.**   We instantiate $\mathcal{T}_{\text{trigger}}$ as a lightweight LoRA adapter attached to the reasoner $\pi_\theta$. At the decoding step $j$ of the timestep $t$, it receives the sequence of all hidden states, $\mathbf{H}_{t,<j} \in \mathbb{R}^{(j-1)\times d_{\text{model}}}$; conditioned on this context, $\mathcal{T}_{\text{trigger}}$ outputs the action probability $P(d_j = \texttt{INVOKE}) \in [0, 1]$. We use hidden states rather than textual context as input because, following prior work in latent reasoning (Hao et al., 2024; Shen et al., 2025a), latent embeddings retain richer context-sensitive information that would otherwise be lost after softmax decoding.   To avoid excessive computational overhead, we adopt a *sentence-granularity activation* strategy, inspired by recent studies on LLM interpretability (Anthropic, 2025; Chen et al., 2024a), which find that interventions between sentences can more effectively guide LLMs' reasoning path.  Specifically, we define a delimiter token set $\mathcal{D}$ (*e.g.*, commas, periods) and let the trigger act only when the current token falls in $\mathcal{D}$. The invocation decision is computed as:

$$d_j = \text{Bernoulli}(p_j), \; p_j = \begin{cases} 0 & \text{if } z_j \notin \mathcal{D}, \\ \mathcal{T}_{\text{trigger}}(\mathbf{H}_{t,<j}) & \text{if } z_j \in \mathcal{D}, \end{cases} \tag{7}$$

which ensures that $\mathcal{T}_{\text{trigger}}$ is invoked only at semantically significant boundaries, preserving decoding efficiency. We validate that `MemGen` does not incur excessive inference delay in Appendix E.3.3.

**Training Recipe.**   The memory trigger is trained via reinforcement learning, motivated by the need to balance two competing desiderata: ensuring that critical latent memories are invoked to improve task performance, while avoiding unnecessary or spurious invocations that could disrupt reasoning or incur computational overhead.  Given a batch of seen tasks $\mathcal{H} = \{(x_i, \tau_i)\}_{i=1}^N$, the frozen reasoner $\pi_\theta$ generates candidate trajectories while the memory weaver $\mathcal{W}_{\text{weaver}}$ remains fixed. At each activated step, the trigger selects an action $\tilde{d}_j \in \{\texttt{INVOKE}, \texttt{SKIP}\}$ and receives a reward $r(\tau_i)$ reflecting the quality of the resulting trajectory with respect to the task objective. To encourage sparse yet strategically critical memory invocation, we introduce a *reward-adaptive penalty*, which discourages unnecessary activations while preserving essential ones, into the objective:

$$\max_\phi \; \mathbb{E}_{\tau_i \sim \pi_\theta, \tilde{\mathbf{d}} \sim \mathcal{T}_{\text{trigger}}^\phi} \left[ R(\tau_i) - \lambda \sum_{i,j} \max(0, \tilde{d}_{i,j} - \bar{p}) \right], \tag{8}$$

where $\bar{p}$ is computed as the mean activation probability across high-reward trajectories, *i.e.*, those with reward exceeding the batch median:

$$\bar{p} = \frac{1}{|\mathcal{H}_{\text{high}}|} \sum_{i \in \mathcal{H}_{\text{high}}} \frac{1}{|\tau_i|} \sum_j \tilde{d}_{i,j}, \; \mathcal{H}_{\text{high}} = \{i : R(\tau_i) \geq \text{median}_k(R(\tau_k))\}, \tag{9}$$

where ensures that $\mathcal{T}_{\text{trigger}}$ learns to invoke memory selectively at key decision points, maximizing task reward while maintaining computational efficiency.

## 4.3   MEMORY WEAVER: SYNTHESIZING AND INSERTING LATENT MEMORY

In this section, we elaborate on the weaver $\mathcal{W}_{\text{weaver}}$, the memory carrier within the `MemGen` framework. When the agent assimilates new experiences, this information is exclusively internalized into the parameters of $\mathcal{W}_{\text{weaver}}$, leaving the core reasoner $\pi_\theta$ entirely unmodified. At junctures where the reasoner requires experiential support, a context-dependent *hook* activates the weaver to synthesize and externalize pertinent knowledge as a usable *memory*. To be more specific, recall from Equation (5) that after the $\mathcal{T}_{\text{trigger}}$ signals the need for memory at step $j$, $\mathcal{W}_{\text{weaver}}$ accepts $\mathbf{H}_{t,<j}$ (as the *hook*) and generates a latent token sequence $\mathbf{M}_t$ (as the *memory*) for $\pi_\theta$.

**Instantiation.**    We instantiate $\mathcal{W}_{\text{weaver}}$ using anthoer LoRA adapter attached to $\pi_\theta$. Formally, given the incoming hook $\mathbf{H}_{t,<j} \in \mathbb{R}^{(j-1)\times d_{\text{model}}}$, the weaver outputs a latent memory matrix: $\mathbf{M}_t = \mathcal{W}_{\text{weaver}}^{\theta'}(\mathbf{H}_{t,<j}) \in \mathbb{R}^{K \times d_{\text{model}}}$, where $K$ denotes the fixed length of the latent memory sequence and $\theta'$ are the trainable LoRA parameters. The synthesized $\mathbf{M}_t$ is first projected through a linear layer to align it with the reasoner's token-embedding space, and is then prepended to the current hidden states of $\pi_\theta$ to guide subsequent token generation, as described in Equation (6).

**Training Recipe.**    The training of $\mathcal{W}_{\text{weaver}}$ proceeds over a batch of past trajectories $\mathcal{H} = \{(x_i, \tau_i)\}_{i=1}^N$. Distinct from conventional agent tuning, which directly integrates experiential data into the parameters of $\pi_\theta$ (Chen et al., 2025; Yin et al., 2024), MemGen internalizes experiential knowledge solely into $\mathcal{W}_{\text{weaver}}$, which ensures that $\pi_\theta$'s general capabilities remain intact.

Crucially, this separation makes MemGen agnostic to optimization strategies and compatible with diverse LLM backbones. Whether employing supervised fine-tuning (SFT) or RL-based objectives such as GRPO or DAPO, the weaver can be updated under a unified goal: optimizing the generation process of latent memory so as to maximize downstream reward. Formally, let $\Pi_\theta^{\mathcal{W}_{\theta'}, \mathcal{T}}(\cdot \mid x)$ denote the process of rolling out a trajectory for a task $x$ by $\pi_\theta$ in conjunction with weaver $\mathcal{W}_{\theta'}$ and trigger $\mathcal{T}$. Given a reward functional $R$, the objective updates only $\theta'$ by maximizing the expected reward:

$$\max_{\theta_{\text{lora}}} \; \mathbb{E}_{(x_i, \tau_i) \sim \mathcal{H}} \; \mathbb{E}_{\tau \sim \Pi_\theta^{\mathcal{W}_{\theta'}, \mathcal{T}}(\cdot | x_i)} \big[ R(x_i, \tau) \big], \tag{10}$$

where the gradients from $R$ are propagated solely to $\theta'$, thereby equipping the weaver to supply precisely the memories that improve end-to-end performance without altering $\pi_\theta$. Equation (10) enables $\mathcal{W}_{\text{weaver}}$ to absorb diverse experiential signals and externalize them as dynamic, context-sensitive latent memories, independent of the architectural or training paradigm of the base reasoner.

In practice, we first train the memory weaver using a random inserter as a lightweight surrogate for the trigger, and then freeze the trained weaver and proceed to train the trigger. For a complete description of the training procedure, please refer to Appendix D.1.

**Integration with Retrieval-based Memory.**    Although the memory generation above primarily draws on the weaver's parametric knowledge, it can be combined with external memory sources. When triggered, any retrieval-based system (*e.g.*, MemoryBank, ExpeL) can provide textual memory, which is merged with the hook $\mathbf{H}_{t,<j}$ and fed into $\mathcal{W}$ to produce latent memory. This allows $\mathcal{W}$ to integrate internal knowledge and external information, supplying the reasoner with richer memory support. Implementation details and results are placed in Appendix F.

## 5    EXPERIMENTS

In this section, we conduct extensive experiments to answer the following research questions:
- (**RQ1**) Can MemGen surpass both parametric and retrieval-based memory?
- (**RQ2**) Is the memory learnt by MemGen generalizable across task domains? And why?
- (**RQ3**) Can MemGen facilitate continual learning and mitigate catastrophic forgetting?
- (**RQ4**) Does MemGen implicitly evolve human-like memory hierachy?

### 5.1    EXPERIMENTAL SETUP

**Evaluation and Benchmarks.**    Our evaluation covers nine datasets from five domains, including ❶ *web search*: TriviaQA (Joshi et al., 2017) and PopQA (Mallen et al., 2023); ❷ *embodied action*: ALFWorld (Shridhar et al., 2021); ❸ *math reasoning*: AQuA (Ling et al., 2017), GSM8K (Cobbe et al., 2021), and MATH (Hendrycks et al., 2021); ❹ *scientific reasoning*: GPQA (Rein et al., 2023); and ❺ *coding*: KodCode (Xu et al., 2025d) and BigCodeBench (Jain et al., 2024).

**Baselines.**    We compare MemGen against twelve baselines, categorized into four groups: **(I) Prompt-based methods**: Vanilla model, CoT (Wei et al., 2023); **(II) Parametric memory**, where experiential knowledge directly modifies model parameters via: SFT, GRPO (DeepSeek-AI et al., 2025), REINFORCE (Williams, 1992), REINFORCE++ (Hu et al., 2025a), Agent-FLAN (Chen et al., 2024b); **(III) Retrieval-based memory**, where processing tasks sequentially and storing the experiences in an external database, represented by MemoryBank (Zhong et al., 2023), ExpeL (Zhao et al., 2024), Agent Workflow Memory (AWM) (Wang et al., 2024c); and **(IV) Latent computation**, where leveraging latent tokens as carriers of experiential knowledge, including SoftCoT (Xu et al., 2025c) and Co-processor (Liu et al., 2024).

Table 1: Results on **SmolLM3-3B** and **Qwen3-8B**. All values represent the performance metric for each task (e.g., accuracy %). We highlight the  best  and  second best  results.

| Backbone | Method | ALFWorld | TrivialQA | PopQA | KodCode | BigCodeBench | GPQA | GSM8K | MATH |
|---|---|---|---|---|---|---|---|---|---|
| SmolLM3-3B | Vanilla | 18.96 | 10.47 | 8.23 | 37.05 | 35.96 | 9.35 | 47.63 | 16.22 |
| | CoT | 17.60 | 12.88 | 9.95 | 38.45 | 39.42 | 20.70 | 58.91 | 56.33 |
| | SFT | 32.36 | 55.25 | 37.22 | 59.25 | 40.79 | 19.70 | 63.48 | 45.65 |
| | GRPO | 55.35 | 65.88 | 45.16 | 68.48 | 72.44 | 22.73 | 80.03 | 61.23 |
| | REINFORCE | 53.13 | 63.20 | 46.81 | 65.53 | 67.14 | 23.44 | 82.03 | 58.75 |
| | REINFORCE++ | 53.95 | 63.20 | 44.10 | 65.90 | 68.80 | 22.73 | 81.50 | 59.89 |
| | Agent-FLAN | 34.00 | 56.70 | 39.50 | 56.80 | 37.20 | 17.80 | 59.60 | 36.84 |
| | ExpeL | 36.18 | 46.20 | 28.16 | 51.14 | 40.22 | 15.15 | 56.23 | 38.11 |
| | MemoryBank | 32.80 | 43.30 | 25.81 | 44.50 | 31.80 | 10.20 | 58.30 | 43.53 |
| | AWM | 40.50 | 49.80 | 29.60 | - | - | - | - | - |
| | SoftCoT | 35.03 | 50.38 | 34.90 | 59.20 | 39.10 | 17.22 | 56.34 | 44.62 |
| | Co-processor | 38.36 | 53.28 | 38.96 | 56.25 | 45.40 | 20.10 | 57.60 | 38.81 |
| | MemGen $_{\mathrm{SFT}}$ | 50.60 | 68.13 | 42.34 | 62.65 | 42.99 | 26.75 | 70.42 | 57.44 |
| | MemGen $_{\mathrm{GRPO}}$ | 63.60 | 79.30 | 58.60 | 72.85 | 74.24 | 25.20 | 83.47 | 63.65 |
| Qwen3-8B | Vanilla | 58.93 | 52.18 | 34.13 | 49.10 | 33.33 | 38.18 | 89.48 | 79.82 |
| | CoT | 57.10 | 53.80 | 33.20 | 51.25 | 35.59 | 35.15 | 87.67 | 78.24 |
| | SFT | 83.59 | 74.55 | 51.12 | 64.75 | 41.33 | 40.33 | 90.76 | 81.35 |
| | GRPO | 85.60 | 76.15 | 58.90 | 73.35 | 70.24 | 39.54 | 92.30 | 83.54 |
| | REINFORCE | 82.10 | 75.22 | 57.96 | 72.11 | 70.20 | 37.12 | 91.25 | 83.27 |
| | REINFORCE++ | 84.80 | 75.90 | 58.30 | 72.90 | 71.88 | 37.68 | 91.90 | 85.24 |
| | Agent-FLAN | 80.32 | 70.32 | 50.08 | 62.99 | 43.40 | 39.50 | 87.60 | 80.05 |
| | ExpeL | 78.97 | 65.54 | 40.33 | 57.20 | 34.23 | 35.15 | 86.20 | 77.40 |
| | MemoryBank | 70.41 | 60.56 | 41.60 | 56.39 | 40.61 | 35.66 | 90.35 | 80.35 |
| | AWM | 80.33 | 69.30 | 43.69 | - | - | - | - | - |
| | SoftCoT | 75.60 | 59.42 | 39.42 | 63.28 | 38.27 | 39.60 | 86.30 | 76.23 |
| | Co-processor | 73.28 | 61.42 | 45.55 | 64.90 | 42.19 | 39.15 | 76.23 | 79.20 |
| | MemGen $_{\mathrm{SFT}}$ | 85.82 | 77.22 | 54.65 | 66.15 | 40.35 | 43.23 | 91.25 | 83.30 |
| | MemGen $_{\mathrm{GRPO}}$ | 90.60 | 80.65 | 62.30 | 76.16 | 75.56 | 40.24 | 93.20 | 88.24 |

**Implementation Details.** We select LLM backbones of varying sizes and sources, including `Qwen-2.5-1.5B` (Yang et al., 2024a), HuggingFace's `SmolLM3-3B` (HuggingFace, 2025), and `Qwen3-8B` (Yang et al., 2025). The length of each latent memory sequence $K$ is set among $\{2, 4, 8\}$. As described in Equation (10), `MemGen` does not rely on a specific optimization algorithm, so we implement two variants: `MemGen` $_{\mathrm{SFT}}$ and `MemGen` $_{\mathrm{GRPO}}$, in which the weaver is updated using SFT and GRPO signals. Details on these variants are provided in Appendix C. More training setup and parameter configurations are listed in Appendix D.

## 5.2 MAIN RESTULS

**[For RQ1] `MemGen` provides high-performing memory across domains.** As shown in Tables 1 and 4, existing baselines exhibit clear limitations in cross-domain adaptivity. Retrieval-based memories (*e.g.*, ExpeL, MemoryBank, AWM) occasionally surpass parazmetric tuning in embodied action; for instance, AWM reaches 36.18% on ALFWorld with `SmolLM3-3B`, exceeding SFT by 3.15%. Yet their effectiveness deteriorates on reasoning-intensive tasks: ExpeL achieves only 8.12% on GPQA+`Qwen2.5-1.5B`, and even underperforms the vanilla model by 6.9% on TriviaQA, underscoring its heavy reliance on backbone capacity. Parametric finetuning methods display the opposite tendency: they excel in structured domains such as code generation, where REINFORCE++ reaches 63.33% on KodCode with `Qwen2.5-1.5B`, but remain weak in knowledge-intensive reasoning, with GPQA below 14%. In contrast, `MemGen` consistently advances performance across all domains. For example, on ALFWorld+`SmolLM3-3B`, `MemGen` $_{\mathrm{SFT}}$ and `MemGen` $_{\mathrm{GRPO}}$ attain 50.60% and 63.60%, improving over vanilla by 31.64% and 44.64%, respectively. Similar gains appear with the larger `Qwen3-8B`, where `MemGen` $_{\mathrm{GRPO}}$ achieves +27.06% on KodCode and +28.17% on PopQA, surpassing GRPO by up to 3.4%. Overall, the dynamic memory insertion of `MemGen` delivers substantial improvements across diverse task domains.

**[For RQ2] `MemGen` Exhibits Strong Cross-Domain Generalization.** To evaluate whether the memory learned by `MemGen` can transfer across tasks, we train `MemGen` on one dataset and test it on several others. We include two out-of-domain datasets, ScienceWorld (Wang et al., 2022) and FEVER (Thorne et al., 2018), to further probe this. As shown in Figures 3, 9 and 10, baselines such as SFT and MemoryBank achieve gains within their training domains (*e.g.*, on ALFWorld, SFT +14.1% and MemoryBank +5.4% compared with vanilla), yet fail to generalize, with performance

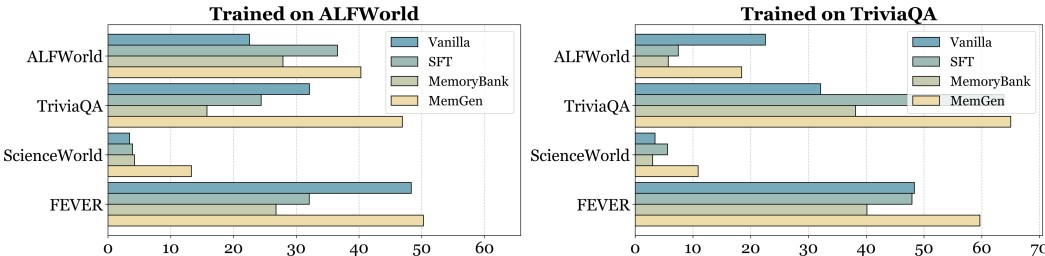

Figure 3: The generalization study of `MemGen`. We train `MemGen` SFT on one dataset (ALFWorld or TriviaQA) and evaluate it on four datasets (TriviaQA, ALFWorld, ScienceWorld, and FEVER).

dropping sharply on FEVER by $16.2\%$. In contrast, `MemGen` not only attains substantial in-domain improvements ($24.55\% \rightarrow 58.16\%$ on KodCode, Figure 10), but also exhibits effective transfer: when trained on KodCode, performance on MATH rises from $36.6\% \rightarrow 54.2\%$. Having empirically validated `MemGen`'s generalizability, a question naturally arises: what underlies this capability?

**[For RQ2] The Memory Trigger Intelligently Determines When to Activate Memory Insertion, Mitigating Domain Conflict.** After training `MemGen` on GSM8K, we evaluate 150 samples each from GSM8K, KodCode, and GPQA, visualizing the frequency with which the memory trigger invoked the memory weaver at each relative position in the model output. We observe that the invocation frequency varies across domains and correlates directly with performance in Figure 9: GSM8K exhibits the largest improvement ($+19.64\%$) and maximal invocations, GPQA achieves moderate gains ($+6.06\%$) with medium invocations, and KodCode shows the smallest improvement ($+3.1\%$) with the fewest invo-

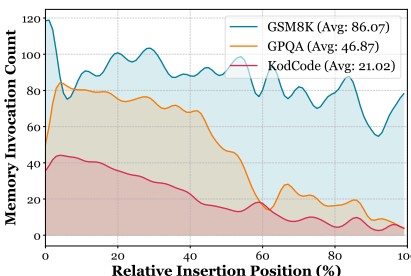

Figure 4: Memory invocation frequency across benchmarks at inference (trained on `MemGen` SFT+Qwen3-8B +GSM8K).

cations. This indicates that `MemGen` autonomously assesses, based on task-specific context, when memory insertion will be beneficial, invoking the weaver less frequently in unfamiliar domains.

**[For RQ3] `MemGen` Effectively Mitigates Catastrophic Forgetting.** In Table 5, we sequentially train on four datasets and evaluate on all benchmarks after each stage, where `MemGen` exhibits stronger knowledge retention ability compared to baseline methods. For example, unlike SFT which primarily improves performance on the most recent task ($54.10\%$ on KodCode but only $2.53\%$ on GPQA), `MemGen` demonstrates more balanced cross-task generalization, attaining $38.43\%$ on AQuA and $21.72\%$ on GPQA after GSM8K training. Finally, it mitigates forgetting on earlier tasks, preserving $40.34\%$ on AQuA following KodCode training compared to $27.14\%$ for ExpeL and $28.61\%$ for SFT, indicating a more stable continual learning ability. More analysis is in Appendix E.1.

## 5.3 Framework Analysis

Having established the expressive capabilities of `MemGen`, we further investigate its underlying mechanisms: what do the learned latent memories look like? Do they have specialized functions?

**[For RQ4] The Latent Memory Is Machine-Native and Human-Unreadable.** We first visualized the latent memory sequences learned by `MemGen` across different datasets using t-SNE in Figures 5 and 11. As shown in Figure 5 (*Left*), sequences from distinct domains form separate distributions, with related domains clustering closely (*e.g.*, KodCode and BigCodeBench, GSM8K and MATH). Examining latent memories within the same dataset, we observed pronounced clustering patterns (as shown in Figure 5 (*Middle* and *Right*)). To explore potential commonalities within these clusters, we forcibly decoded the latent tokens. Although the decoded sequences are not human-readable, they exhibit intriguing regularities: many tokens within a cluster share structural conventions. For example, Cluster 0 in TriviaQA frequently follows the pattern "`[...]SOC`", whereas Cluster 3 in GSM8K often adopts the format "`[...]_pick`". A large corpus of latent memory tokens is provided in Appendix G. Despite these sequences being machine-native and human-unreadable, we further investigate whether their underlying semantics can be interpreted.

**[For RQ4] `MemGen` Implicitly Learns a Human-like Memory Hierarchy.** To uncover the functional roles of different latent memory clusters, we conducted a post-hoc intervention study. Following the taxonomy from (Song et al., 2025), we study eight distinct types of agent failure, including

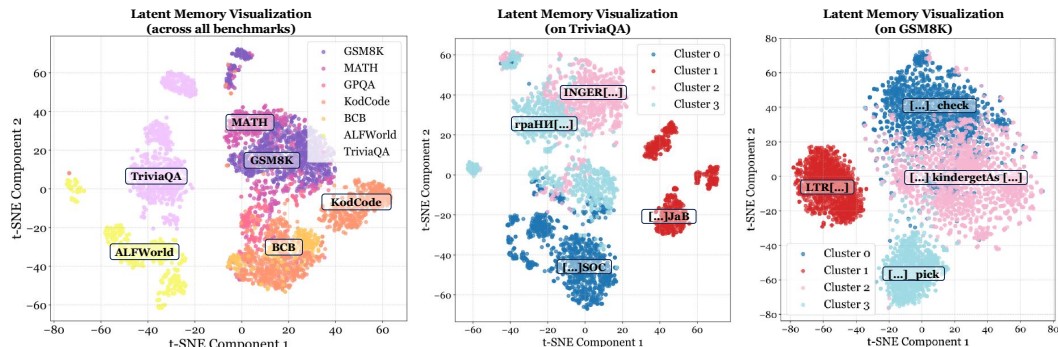

Figure 5: (*Left*) t-SNE visualization of latent memories generated by `MemGen` +`Qwen3-8B` across datasets; (*Middle* and *Right*) Latent memory visualization within the TriviaQA and GSM8K datasets, clustered using $K$-means. The text at each cluster center represents the common pattern shared by many memory sequences in the cluster, such as Cluster 0 in GSM8K, where many sequences end with "`_check`".

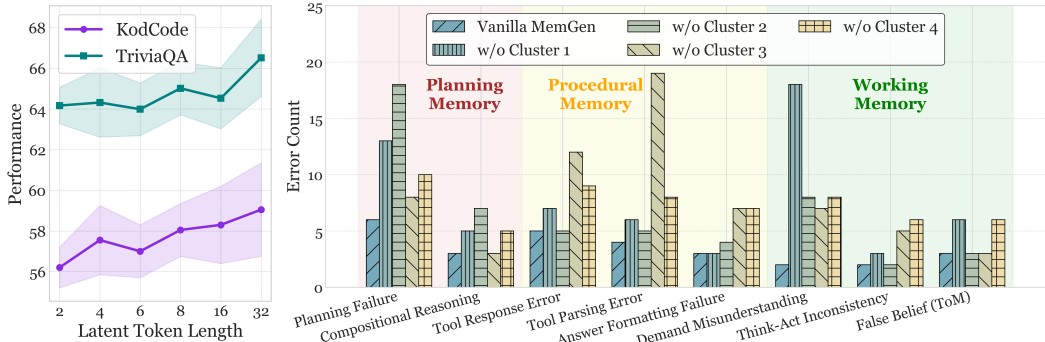

Figure 6: (*Left*) Parameter sensitivity analysis on the latent memory length $K$; (*Right*) Effects of selectively removing latent memory clusters on different agent failure modes on the TriviaQA dataset.

planning errors, tool response/parsing failures, answer formatting mistakes, *etc*, providing a structured framework to assess how memory influences performance. During evaluation, we selectively removed latent tokens close to a specific cluster while keeping others intact, measuring the resulting changes in these failure modes. Details on (1) the visualization process, (2) failure mode annotation, and (3) token filtration are in Appendix H. As shown in Figure 6 (*Right*), distinct memory clusters exhibit varying influence on failure modes and can be mapped to different memory functions:

- **Planning Memory** supports high-level task planning and strategic reasoning. Removal of Cluster 2 substantially increases planning and compositional reasoning failures, indicating that this cluster is crucial for guiding the LLM agent's decision-making and sequencing of reasoning steps.
- **Procedural Memory** captures task-specific operational knowledge, such as tool usage and formatting ability. Cluster 3 corresponds to this role, as its removal leads to a marked increase in tool response errors, parsing failures, and answer formatting mistakes.
- **Working Memory** manages the retention and effective use of prior context to maintain reasoning consistency. Clusters 1 and 4 contribute to this function: for instance, removing Cluster 1's memory tokens results in more frequent task misunderstandings and think-act inconsistency.

Nevertheless, these memory clusters are not entirely independent: for example, removing Cluster 1 also negatively affects planning ability, indicating that these memory faculties interact and jointly enable the LLM to leverage past experience effectively. This analysis reveals that `MemGen` spontaneously organizes latent memory into a structured, human-like hierarchy.

**Ablation Study & Sensitivity Analysis.** We conduct a sensitivity analysis on the length of the latent memory sequence $K$, as shown in Figure 6 (*Left*). It can be observed that as the latent token length increases from $2 \rightarrow 32$, `MemGen`'s performance correspondingly improves, likely reflecting the expanded memory capacity. We then perform an ablation study on the memory trigger module in Table 6, demonstrating the necessity of a dedicatedly trained trigger for effective memory invocation. Furthermore, we analyze different training paradigms of the memory weaver in Table 7. For additional results and discussions, please refer to Appendix E.3.

**Efficiency Analysis.** To confirm that the memory insertion process of `MemGen` does not introduce significant inference overhead, we show in Appendix E.3.3 that, while achieving up to $57.66\%$ performance improvement over vanilla LLMs, the per-query inference delay remains consistently below the baseline, ranging from $24\%$ to $94\%$ of the vanilla LLM latency. This clearly demonstrates that `MemGen` delivers substantial performance gains without compromising efficiency.

## 6 CONCLUSION

In this work, we introduced `MemGen`, a dynamic and generative memory framework designed for LLM Agents. By interleaving reasoning with memory synthesis through a reinforcement-learned memory trigger and a generative memory weaver, `MemGen` transcends the limitations of parametric and retrieval-based paradigms. Extensive experiments showcase substantial performance gains, robust cross-domain generalization, strong continual learning ability, and `MemGen`'s explicitly modeled memory hierarchy (*i.e.*, planning, procedural, and working memory). These results suggest a promising path toward self-evolving LLM agents capable of fluid and reconstructive intelligence.

## ETHICS STATEMENT

This work presents a latent memory architecture designed for LLM agents. All experiments are conducted on publicly available academic benchmarks across reasoning, scientific problem-solving, and embodied tasks, without deployment in real-world decision-making or safety-critical applications. Therefore, we believe this research does not pose direct ethical risks.

## REPRODUCIBILITY STATEMENT

We aim to ensure the reproducibility of our work by releasing an anonymous repository linked in the abstract, as well as detailing experimental settings (hyperparameters, LLM backbones, *etc*.) in §5.1 and appendix D.

## CONTRIBUTIONS

G. Zhang and M. Fu contributed equally to this work. G. Zhang led the initial framework implementation, manuscript preparation, and experimental analysis. M. Fu made key contributions to improving the efficiency of MemGen and conducted large-scale experiments and visualization. S. Yan is the corresponding author.

## ACKNOWLEDGEMENTS

This research was supported in part by the NUS Start-up Grant (A-0010106-00-00) and by the National Natural Science Foundation of China under Grant No. 62320106007.

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

## A  USE OF LARGE LANGUAGE MODELS

In preparing this work, we employed large language models (LLMs) to assist with: polishing the writing, conducting literature reviews, and creating visualizations.

## B  ADDITIONAL RELATED WORKS

**LLM Decoding & RL.**   Two additional topics that relate to our work are LLM decoding and reinforcement learning (RL). From the **decoding perspective**, `MemGen` dynamically generates and inserts latent tokens, which shares similarity with speculative decoding where a drafter model receives the current decoding context and produces subsequent drafted tokens (Cai et al., 2024; Fu et al., 2024; Li et al., 2025b; Goel et al., 2025). . However, these methods primarily aim to accelerate LLM inference, whereas `MemGen` focuses on leveraging latent states as effective carriers of memory. From the **RL perspective**, `MemGen` employs rule-based RL to train the memory trigger, which is closely related to reinforcement learning with variable reward (RLVR), including GRPO from DeepSeek-R1 (DeepSeek-AI et al., 2025) and its various derivatives (Qian et al., 2025; Wu et al., 2025a; Wei et al., 2025; Fu et al., 2025). While there exist efforts combining RL with agent memory, to our knowledge, most do not address self-improving memory; for example, MemAgent (Yu et al., 2025) and MEM1 (Zhou et al., 2025) focus on handling long-context inputs rather than evolving memory mechanisms.

**Latent Memory**   We further extend our discussion by incorporating related work on latent memory mechanisms: R3Mem (Wang et al., 2025a) introduces a reversible, hierarchical memory architecture that integrates efficient long-term information retention with faithful retrieval, achieving state-of-the-art performance in long-context modeling, retrieval-augmented generation, and prolonged interactive tasks.

## C  OPTIMIZATION ALGORITHM DETAILS

In this section, we provide a detailed exposition of the optimization algorithms for training the memory weaver, $\mathcal{W}_{\text{weaver}}$, as mentioned in §4.3. The core principle, as established in Equation (10), is to update only the weaver's parameters, denoted as $\theta'$, while keeping the reasoner $\pi_\theta$ frozen. This modularity allows `MemGen` to be compatible with various optimization paradigms. We detail the specific implementations for Supervised Fine-Tuning (SFT) and a reinforcement learning approach, Group Relative Policy Optimization (GRPO).

### C.1  COMBINING `MemGen` WITH SFT

The objective of Supervised Fine-Tuning is to train the memory weaver to generate latent memories that guide the frozen reasoner $\pi_\theta$ to replicate the behavior observed in a dataset of high-quality demonstration trajectories. We leverage the provided history of past experiences, $\mathcal{H} = \{(x_i, \tau_i^*)\}_{i=1}^N$, where each $\tau_i^*$ is treated as an expert demonstration.

Each expert trajectory $\tau_i^*$ consists of a sequence of states and actions, $\tau_i^* = (s_{i,0}, a_{i,0}^*, s_{i,1}, a_{i,1}^*, \dots)$. Each expert action $a_{i,t}^*$ is a sequence of tokens, $a_{i,t}^* = (\mathbf{z}_{i,t,1}^*, \mathbf{z}_{i,t,2}^*, \dots, \mathbf{z}_{i,t,L_t}^*)$. The goal is to maximize the conditional log-likelihood of generating these expert token sequences.

During the training of the weaver, both the reasoner $\pi_\theta$ and the memory trigger $\mathcal{T}$ are held fixed. At each token generation step $j$ where the trigger activates, the weaver $\mathcal{W}_{\theta'}$ takes the reasoner's hidden states $\mathbf{H}_{i,t,<j}$ as input and produces a latent memory $\mathbf{M}_{i,t}$. The reasoner then generates the next token conditioned on this memory. The SFT objective is to adjust the weaver's parameters $\theta'$ to maximize the probability of the ground-truth token $\mathbf{z}_{i,t,j}^*$.

Formally, the optimization problem is to minimize the negative log-likelihood of the expert trajectories, averaged over the dataset $\mathcal{H}$. The loss function for the weaver's parameters $\theta'$ is defined as:

$$\mathcal{L}_{\text{SFT}}(\theta') = -\mathbb{E}_{(x_i, \tau_i^*) \sim \mathcal{H}} \left[ \sum_{t=0}^{T_i-1} \sum_{j=1}^{L_t} \log \pi_\theta(\mathbf{z}_{i,t,j}^* \mid s_{i,t}, \mathbf{z}_{i,t,<j}^*, \mathbf{M}_{i,t,j}) \right], \qquad (11)$$

where the latent memory $\mathbf{M}_{i,t,j}$ is synthesized by the weaver at that specific step:

$$\mathbf{M}_{i,t,j} = \mathcal{W}_{\theta'}(\mathbf{H}_{i,t,<j}). \qquad (12)$$

Note that the generation of $\mathbf{M}_{i,t,j}$ only occurs if the fixed trigger $\mathcal{T}$ determines an invocation is needed at step $j$. In steps where no memory is invoked, the conditioning term $\mathbf{M}_{i,t,j}$ is omitted. The

gradients are computed exclusively with respect to the weaver's parameters $\theta'$ and used to update them via gradient descent:

$$\theta' \leftarrow \theta' - \eta \nabla_{\theta'} \mathcal{L}_{\text{SFT}}(\theta'), \tag{13}$$

where $\eta$ is the learning rate. Through this process, the memory weaver learns to synthesize latent memories that effectively steer the frozen reasoner's generative process to align with the training data's behavior, thereby internalizing the knowledge from the demonstration data without corrupting the general capabilities of the core LLM.

## C.2  Combining `MemGen` with GRPO

The memory weaver can also be trained using a reinforcement learning objective. We specifically adapt the GRPO algorithm. The training process begins by sampling a batch of tasks from the experience history $\mathcal{H}$. For each task $x_i$, we use the policy $\Pi_\theta^{\mathcal{W}_{\theta'}, \mathcal{T}}$ (composed of the frozen reasoner $\pi_\theta$ and the current memory weaver $\mathcal{W}_{\theta'}$) to generate a group of $K$ distinct trajectories, denoted as $\mathcal{G}_i = \{\tau_{i,1}, \tau_{i,2}, \ldots, \tau_{i,K}\}$. Each trajectory is generated by the agent's interaction with the environment and results in a final reward $R(\tau_{i,k})$, evaluated by the reward function. Going forward, GRPO computes a group-relative baseline by averaging the rewards of all trajectories within the group $\mathcal{G}_i$:

$$\bar{R}(\mathcal{G}_i) = \frac{1}{K} \sum_{k=1}^{K} R(\tau_{i,k}). \tag{14}$$

The advantage for a specific trajectory $\tau_{i,k}$ is then its reward relative to this baseline:

$$A(\tau_{i,k}) = R(\tau_{i,k}) - \bar{R}(\mathcal{G}_i). \tag{15}$$

This formulation allows the weaver to learn by differentiating between better and worse outcomes within a set of its own generations, promoting policies that produce trajectories with above-average rewards. The final objective function, maximized with respect to the weaver's parameters $\theta'$, is:

$$\mathcal{J}_{\text{GRPO}}(\theta') = \mathbb{E}_{x_i \sim \mathcal{H}, \, \mathcal{G}_i \sim \Pi_\theta^{\mathcal{W}_{\theta'}, \mathcal{T}}} \left[ \frac{1}{K} \sum_{k=1}^{K} A(\tau_{i,k}) \log \Pi_\theta^{\mathcal{W}_{\theta'}, \mathcal{T}}(\tau_{i,k} \mid x_i) - \beta \, \text{KL}\left( \Pi_\theta^{\mathcal{W}_{\theta'}, \mathcal{T}}(\cdot \mid x_i) \,\|\, \Pi_{\text{ref}}(\cdot \mid x_i) \right) \right], \tag{16}$$

where $\Pi_{\text{ref}}$ is a fixed reference policy and $\beta$ is a coefficient controlling the strength of the KL regularization. The gradients are computed only for the weaver's parameters $\theta'$, thus refining its ability to synthesize impactful latent memories while preserving the integrity of the core reasoner.

## C.3  Memory Trigger Training Details

We now detail the training procedure of the memory trigger $\mathcal{T}_{\text{trigger}}^\phi$. During this stage, the reasoner $\pi_\theta$ and the memory weaver $\mathcal{W}_{\theta'}$ remain frozen, and only the trigger parameters $\phi$ are updated. For each task $x_i$ with initial environment state $s_0$, we generate trajectories under the composed policy induced by $\pi_\theta$, $\mathcal{W}_{\theta'}$, and $\mathcal{T}_{\text{trigger}}^\phi$. At each decoding step $t$ and token position $j$ where $z_{t,j} \in \mathcal{D}$, the trigger observes the hidden-state prefix $\mathbf{H}_{t,<j}$ and samples a binary action

$$d_{t,j} \sim \pi_\phi(d \mid \mathbf{H}_{t,<j}) := \text{Bernoulli}(p_{t,j}), \qquad p_{t,j} = \sigma\big(\mathcal{T}_{\text{trigger}}^\phi(\mathbf{H}_{t,<j})\big), \tag{17}$$

where $d_{t,j} = 1$ corresponds to INVOKE. If invoked, the decoder pauses and the weaver synthesizes a latent memory $\mathbf{M}_t$ from $\mathbf{H}_{t,<j}$, after which generation resumes based on the augmented context. Let $\mathcal{I}_i$ denote the set of activated positions in trajectory $\tau_i$, and let $\tilde{\mathbf{d}}_i = \{d_{t,j}\}_{(t,j) \in \mathcal{I}_i}$ be the trigger decisions within that rollout. After completing the episode, the environment returns a scalar reward $R(\tau_i)$, which we assign uniformly to all decisions in $\tilde{\mathbf{d}}_i$.

To discourage excessively frequent invocation, we introduce a sparsity-inducing penalty. For a group of rollouts generated from the same task $x_i$, we identify the high-reward subset

$$\mathcal{H}_{\text{high}} = \{i : R(\tau_i) \geq \text{median}_k R(\tau_k)\}, \tag{18}$$

and compute the reference activation level

$$\bar{p} = \frac{1}{|\mathcal{H}_{\text{high}}|} \sum_{i \in \mathcal{H}_{\text{high}}} \frac{1}{|\mathcal{I}_i|} \sum_{(t,j) \in \mathcal{I}_i} d_{t,j}, \tag{19}$$

which estimates the typical invocation frequency among successful trajectories. The trigger is trained to maximize the reward while penalizing activations above $\bar{p}$:

$$\max_{\phi} \ \mathbb{E}_{\tau_i \sim \pi_\theta, \, \tilde{\mathbf{d}}_i \sim \mathcal{T}_{\text{trigger}}^\phi} \left[ R(\tau_i) - \lambda \sum_{(t,j) \in \mathcal{I}_i} \max\left(0, d_{t,j} - \bar{p}\right) \right], \tag{20}$$

where $\lambda > 0$ controls the sparsity regularization strength.

We optimize this objective following a GRPO-style procedure. For each task prompt, we generate a group of $G$ trajectories $\{\tau_i^{(g)}\}_{g=1}^G$ (we set $G = 8$ in practice; Table 3). We compute per-trajectory rewards $R(\tau_i^{(g)})$ and construct normalized advantages

$$A_i^{(g)} = \frac{R(\tau_i^{(g)}) - \mu_i}{\sigma_i}, \qquad \mu_i = \tfrac{1}{G} \sum_{g=1}^G R(\tau_i^{(g)}), \qquad \sigma_i = \sqrt{\tfrac{1}{G} \sum_{g=1}^G (R(\tau_i^{(g)}) - \mu_i)^2},$$

thus measuring each rollout's performance relative to the group. The trigger parameters are then updated by maximizing a GRPO-style surrogate objective:

$$\max_{\phi} \ \frac{1}{G} \sum_{g=1}^G \sum_{(t,j) \in \mathcal{I}_i^{(g)}} \log \pi_\phi\big(d_{t,j}^{(g)} \mid \mathbf{H}_{t,<j}^{(g)}\big) A_i^{(g)} \ - \ \beta \, \mathrm{KL}\big(\pi_\phi \, \| \, \pi_{\phi_{\text{old}}}\big).$$

After each update, we set $\pi_{\phi_{\text{old}}} \leftarrow \pi_\phi$ and repeat. This objective allows the trigger to learn to invoke memory only at critical points that improve group-relative performance while maintaining sparsity.

## D   EXPERIMENTAL DETAILS

### D.1   TRAINING DATASET SETUP

For all datasets except PopQA, we utilize their official training sets to train both the *memory weaver* and *memory trigger*. Specifically, we first train the memory weaver in isolation. Since the trigger module is not yet available at this stage, latent memory tokens are randomly inserted at punctuation marks during training to provide diverse learning signals. Once the memory weaver is adequately trained, we proceed to train the memory trigger, enabling it to learn when to invoke memory (with the memory weaver fixed). For PopQA, which lacks a dedicated training set, we instead leverage the trained model on TriviaQA and directly evaluate on PopQA. The dataset statistics are summarized in Table 2.

Table 2: Dataset Statistics.

| Dataset Name | Train Size | Test Size | Note |
|---|---|---|---|
| ALFWorld | 3.32K | 134 | From AgentBank-ALFWorld |
| TriviaQA | 4.13K | 7.9K | From AgentBank-TriviaQA |
| PopQA | - | 14.3K | - |
| KodCode | 8K | 2K | KodCode-Light-RL-10K |
| BigCodeBench | 912 | 228 | - |
| GPQA | 448 | 198 | GPQA-Diamond for testing |
| GSM8K | 7.47K | 1K | - |
| MATH | 1.6K | 500 | - |

## D.2 PARAMETER CONFIGURATIONS

Table 3: Hyperparameters used in the training of MemGen.

| Settings | Hyperparameters |
|---|---|
| **Training(SFT)** | train_batch_size = 4
learning_rate = 1e-5
epochs = 2
warmup_ratio = 0.1
optim = adamw_torch
schedular = cosine |
| **Training(GRPO)** | rollout_batch_size = 8
train_batch_size = 8
epochs = 2
beta = 0.0
num_iterations = 1
learning_rate = 1e-5
warmup_ratio = 0.1
optim = adamw_torch
schedular = cosine |
| **LoRA** | r = 16
lora_alpha = 32
target_modules = [q_proj, v_proj]
lora_dropout = 0.1
bias = none
task_type = CAUSAL_LM |
| **Optimization** | adam_offload
flash_attn
deepspeed_enable_sleep |

## E EXTRA RESULTS

Table 4: Results on **Qwen2.5-1.5B**. All values represent the performance metric for each task (e.g., accuracy %). We highlight the best and second best results.

| Method | ALFWorld | TrivialQA | PopQA | KodCode | BigCodeBench | GPQA | GSM8K | MATH |
|---|---|---|---|---|---|---|---|---|
| Vanilla | 22.54 | 32.10 | 16.08 | 24.55 | 40.35 | 11.62 | 39.51 | 36.63 |
| CoT | 18.30 | 28.67 | 18.39 | 32.32 | 38.59 | 15.67 | 56.79 | 45.22 |
| SFT | 36.57 | 63.84 | 39.20 | 55.83 | 37.72 | 11.11 | 54.83 | 38.84 |
| GRPO | 43.55 | 68.21 | 43.15 | 62.11 | 70.34 | 15.65 | 68.10 | 47.42 |
| REINFORCE | 43.25 | 66.50 | 41.87 | 60.20 | 67.80 | 12.50 | 67.40 | 46.89 |
| REINFORCE++ | 43.66 | 66.90 | 44.69 | 63.33 | 69.50 | 13.80 | 69.04 | 47.33 |
| Agent-FLAN | 35.80 | 64.28 | 38.90 | 56.21 | 43.83 | 9.35 | 53.02 | 29.82 |
| ExpeL | 28.96 | 25.20 | 20.20 | 31.15 | 39.78 | 8.12 | 45.12 | 38.12 |
| MemoryBank | 27.89 | 38.14 | 22.78 | 37.93 | 35.87 | 13.87 | 47.88 | 30.47 |
| AWM | 30.42 | 55.69 | 32.54 | - | - | - | - | - |
| SoftCoT | 33.07 | 62.22 | 38.78 | 55.13 | 36.10 | 9.31 | 54.50 | 38.55 |
| Co-processor | 35.66 | 64.78 | 40.12 | 56.65 | 38.10 | 12.12 | 57.12 | 37.40 |
| MemGen $_{\text{SFT}}$ | 40.30 | 65.02 | 41.28 | 58.16 | 42.47 | 18.28 | 58.15 | 47.12 |
| MemGen $_{\text{GRPO}}$ | 54.27 | 73.42 | 49.28 | 65.43 | 72.81 | 18.18 | 73.39 | 53.36 |

*(Left margin label: Qwen2.5-1.5B)*

### E.1 CONTINUAL LEARNING RESULT

The results in Table 5 indicate three main findings. First, MemGen exhibits stronger knowledge retention and forward transfer than SFT and ExpeL. For example, when trained on GPQA, MemGen reaches 47.96% on GSM8K and 28.80% on KodCode, surpassing SFT at 45.74% and 18.50% by margins of +2.22% and +10.30%, respectively. Similarly, when trained on KodCode, MemGen maintains 40.34% on AQuA and 20.09% on GPQA, whereas SFT yields 28.61% and 2.53%. Second, MemGen demonstrates more balanced cross-task generalization. Unlike SFT, which primarily improves performance on the most recent task (e.g., 54.10% on KodCode but only 2.53% on GPQA), MemGen achieves competitive results across tasks. After training on GSM8K, it attains 38.43% on AQuA and 21.72% on GPQA, both higher than SFT and ExpeL, suggesting that latent memory captures task-invariant reasoning. Third, MemGen effectively mitigates forgetting on earlier tasks. After sequential training on KodCode, it preserves 40.34% on AQuA compared to 27.14% of ExpeL and 28.61% of SFT, showing stronger robustness to catastrophic forgetting. These findings suggest that MemGen provides a more stable and transferable mechanism for continual learning.

### E.2 TRIGGER FREQUENCY VISUALIZATION

More visualizations of the trigger frequency are displayed in Figures 7 and 8, where we paired `Qwen2.5-1.5B` or `SmolLM3-3B` with the GSM8K dataset and tested on subsets of GSM8K, KodCode, and GPQA (each subset having the same number of samples). We then tallied the frequency of memory trigger `INVOKE` judgments at each relative percentile position in the LLM output.

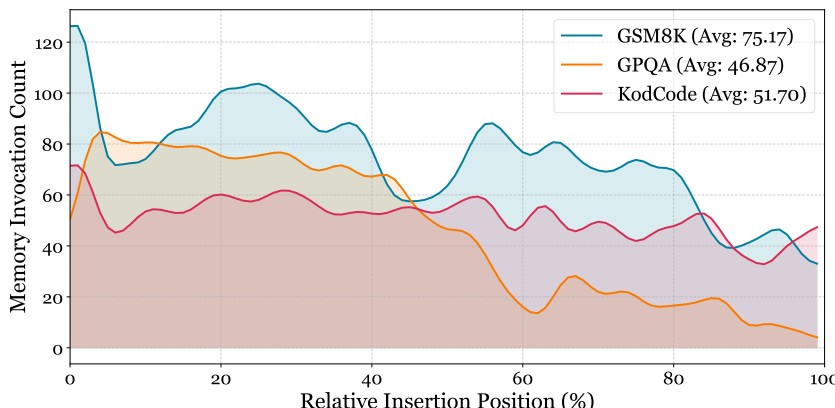

Figure 7: Memory invocation frequency across benchmarks at inference (trained on **MemGen** SFT+`Qwen2.5-1.5B` +GSM8K).

### E.3 FRAMEWORK ANALYSIS

#### E.3.1 ABLATION STUDY

To assess the effectiveness of the proposed memory trigger, we conduct an ablation study comparing different memory invocation strategies, as summarized in Table 6. Specifically, *Random* denotes a naïve baseline where latent memory tokens are inserted at arbitrary token positions with a fixed probability $p$. *All delimiters activated* represents a sentence-level strategy that invokes the memory weaver at every delimiter position without any selection mechanism. Finally, **MemGen**'s *dedicated Trigger* corresponds to our standard approach, where the memory weaver is activated by the trained trigger $\mathcal{T}_{\text{trigger}}$. The results reveal several key observations. First, sentence-level intervention already improves performance compared to random invocation. For instance, activating the weaver at all delimiters yields 17.34%, 56.20%, and 64.15% on GPQA, Kodcode, and TriviaQA, respectively, consistently outperforming all random baselines (e.g., $p = 0.5$ achieves only 16.66%, 52.95%, and 57.28%). This highlights the importance of aligning memory injection with semantic boundaries rather than distributing it across the sequence. More importantly, our trained trigger achieves the best overall performance, reaching 18.28%, 58.16%, and 65.02% on the three benchmarks. This

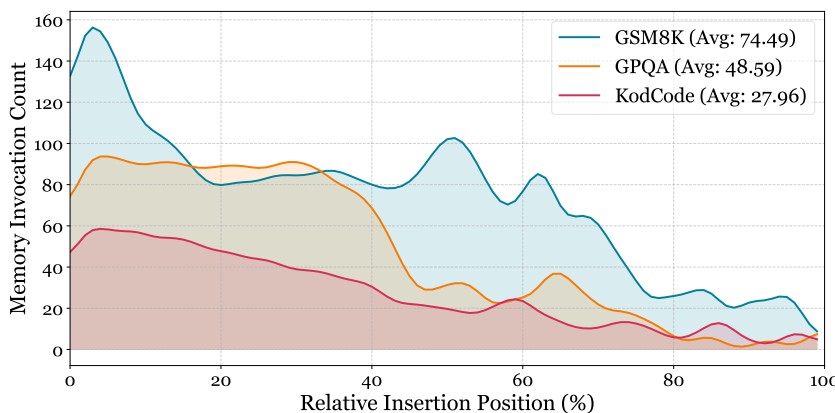

Figure 8: Memory invocation frequency across benchmarks at inference (trained on `MemGen`SFT+`SmolLM3-3B`+GSM8K).

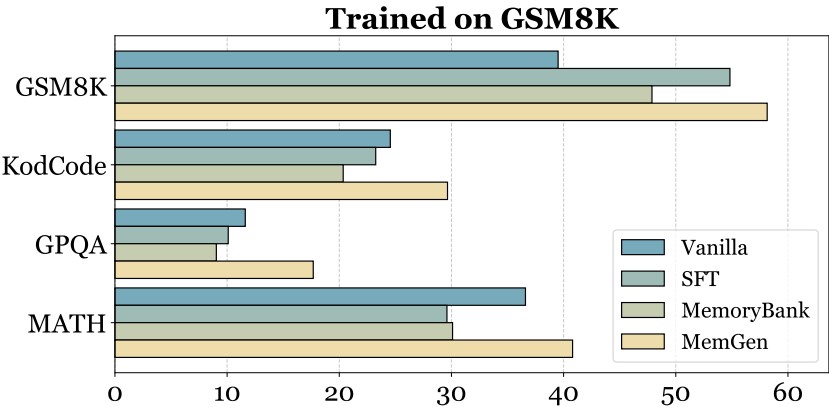

Figure 9: The generalization study of `MemGen`. We train `MemGen` SFT on GSM8K and evaluate it on all four datasets.

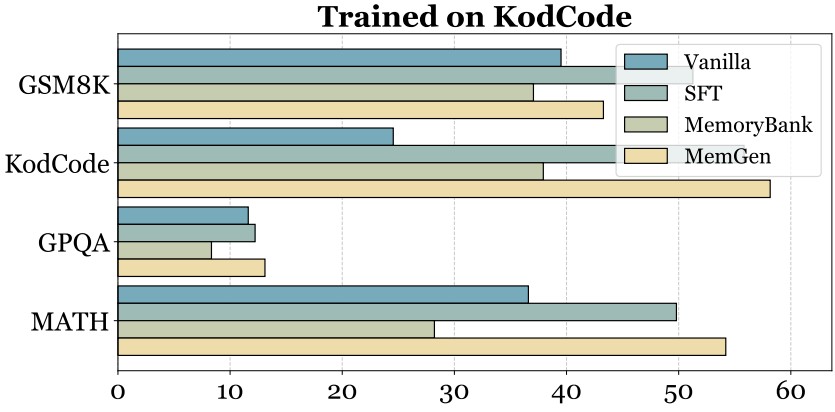

Figure 10: The generalization study of `MemGen`. We train `MemGen` SFT on KodCode and evaluate it on all four datasets.

demonstrates that selective activation, *i.e.*, deciding *when* and *where* to weave in memory, provides the most effective support for reasoning, as it balances memory utility and interference more precisely than coarse-grained alternatives.

Table 5: **Continual learning results** of `Qwen2.5-1.5B-Instruct` across four reasoning and programming datasets (AQuA, GPQA, GSM8K, KodCode). The model is sequentially trained on each dataset (AQuA → GPQA → GSM8K → KodCode), and after each training stage, evaluation is conducted on all four benchmarks.

| Trained On | Method | AQuA | GPQA | GSM8K | KodCode |
|---|---|---|---|---|---|
| | Vanilla | 41.34 | 11.62 | 39.51 | 24.55 |
| AQuA | SFT | 42.52 | 16.67 | **42.10** | 18.20 |
| | ExpeL | 41.73 | 12.67 | 40.16 | 16.30 |
| | MemGen $_{\mathsf{SFT}}$ | **43.31** | **19.70** | 39.80 | **19.55** |
| GPQA | SFT | 38.55 | 17.17 | 45.74 | 18.50 |
| | ExpeL | 37.24 | 14.35 | 42.67 | 15.20 |
| | MemGen $_{\mathsf{SFT}}$ | **39.85** | **20.72** | **47.96** | **28.80** |
| GSM8K | SFT | 33.46 | 13.13 | 52.31 | 19.45 |
| | ExpeL | 34.89 | 12.42 | 48.78 | 13.65 |
| | MemGen $_{\mathsf{SFT}}$ | **38.43** | **21.72** | **55.67** | **19.75** |
| KodCode | SFT | 28.61 | 2.53 | 24.14 | **54.10** |
| | ExpeL | 27.14 | 6.23 | 31.44 | 48.35 |
| | MemGen $_{\mathsf{SFT}}$ | **40.34** | **20.09** | **53.72** | 52.95 |

Table 6: Ablation study of different memory invocation strategies. Random denotes a naïve baseline where latent memory tokens are inserted at arbitrary token positions with a fixed probability $p$. All delimiters activated represents a sentence-level strategy that invokes the memory weaver at every delimiter position without any selection mechanism. Finally, **MemGen**'s dedicated Trigger corresponds to our standard approach, where the memory weaver is activated by the trained trigger $\mathcal{T}_{\mathrm{trigger}}$.

| Memory Invocation Strategy | GPQA | Kodcode | TriviaQA |
|---|---|---|---|
| Random ($p = 0.2$) | 15.66 | 54.55 | 63.55 |
| Random ($p = 0.5$) | 16.66 | 52.95 | 57.28 |
| Random ($p = 0.8$) | 12.63 | 53.60 | 62.22 |
| All delimiters activated | 17.34 | 56.20 | 64.15 |
| **MemGen**'s dedicated Trigger | **18.28** | **58.16** | **65.02** |

### E.3.2 ANALYSIS OF MEMORY WEAVER

We further investigate how the parameterization of the memory weaver influences downstream performance. As shown in Table 7, increasing the number of trainable parameters by adopting a full-parameter SFT setup surely enhances the weaver's memory and learning capacity, leading to higher task performance across benchmarks. Nevertheless, the LoRA-based instantiation already achieves competitive results while retaining high parameter efficiency, demonstrating that even lightweight adaptations can endow the weaver with sufficient capacity to generate effective latent memories.

### E.3.3 EFFICIENCY ANALYSIS

Table 8 reports average inference time and task performance across three benchmarks. Both SFT and **MemGen** drastically reduce per-task inference time compared with vanilla models due to fewer tokens required to reach correct answers. For instance, on KodCode with `Qwen2.5-1.5B`, **MemGen** $_{\mathsf{SFT}}$ completes tasks in 2.94 s, a 75.4% reduction from 11.96 s for vanilla, while improving accuracy by 33.61%. On ALFWorld+`Qwen3-8B`, **MemGen** $_{\mathsf{SFT}}$ adds only 1.6% more time compared with SFT (20.08 s vs 19.76 s) but increases accuracy from 83.59% to 85.82%. These results confirm that **MemGen** efficiently enhances reasoning performance without incurring significant inference delays.

Table 7: Ablation study of the latent weaver. We instantiate `MemGen` $_{\mathsf{SFT}}$'s memory weaver with LoRA of different ranks as well as full-parameter SFT.

| Base LLM: **Qwen2.5-1.5B-Instruct** | GPQA | Kodcode | TriviaQA |
|---|---|---|---|
| LoRA ($r = 4$) | 17.16 | 54.85 | 63.04 |
| LoRA ($r = 6$) | 18.18 | 55.25 | 64.55 |
| LoRA ($r = 8$) | 17.67 | 55.75 | 64.10 |
| LoRA ($r = 16$) | 18.28 | 56.16 | 65.02 |
| LoRA ($r = 24$) | 18.67 | 57.20 | 65.40 |
| LoRA ($r = 32$) | 19.26 | 57.95 | 65.85 |
| Full SFT | 21.21 | 60.00 | 67.10 |

Table 8: Average per-task inference time (seconds) and task performance (accuracy %) across three benchmarks. Performance improvement (%) of `MemGen` $_{\mathsf{SFT}}$ over vanilla LLM is also reported.

| Model & Method | **KodCode** | | **ALFWorld** | | **TriviaQA** | |
|---|---|---|---|---|---|---|
| | Time (s) | Acc (%) | Time (s) | Acc (%) | Time (s) | Acc (%) |
| Qwen2.5-1.5B | | | | | | |
| Vanilla | 11.96 | 24.55 | 21.17 | 22.54 | 2.18 | 32.10 |
| SFT | 2.01 | 55.83 | 10.79 | 36.57 | 1.98 | 63.84 |
| `MemGen` $_{\mathsf{SFT}}$ | 2.94 | 58.16 | 12.94 | 40.30 | 2.05 | 65.02 |
| Improvement over Vanilla | - | +33.61 | - | +17.76 | - | +32.92 |
| SmolLM-3B | | | | | | |
| Vanilla | 13.12 | 37.05 | 34.82 | 18.96 | 4.26 | 10.47 |
| SFT | 3.04 | 59.25 | 12.88 | 32.36 | 3.05 | 55.25 |
| `MemGen` $_{\mathsf{SFT}}$ | 3.48 | 62.65 | 14.69 | 50.60 | 3.16 | 68.13 |
| Improvement over Vanilla | - | +25.60 | - | +31.64 | - | +57.66 |
| Qwen3-8B | | | | | | |
| Vanilla | 16.99 | 49.10 | 55.42 | 58.93 | 8.70 | 52.18 |
| SFT | 7.24 | 64.75 | 19.76 | 83.59 | 5.98 | 74.55 |
| `MemGen` $_{\mathsf{SFT}}$ | 7.56 | 66.15 | 20.08 | 85.82 | 6.25 | 77.22 |
| Improvement over Vanilla | - | +17.05 | - | +26.89 | - | +25.04 |

### E.3.4 ABLATION OF MEMORY WEAVER: PAUSE TOKEN

To further assess the necessity of the memory weaver, we introduce a strong and conceptually aligned baseline based on the *pause token* mechanism (Goyal et al., 2024). Pause tokens are a small set of learned special tokens $\mathcal{P} = \{\texttt{<pause>}_1, \ldots, \texttt{<pause>}_K\}$ that instruct the model to suspend outward generation while continuing to update its internal hidden states. Formally, inserting a pause token $\texttt{<pause>}$ at position $j$ forces the model to compute a new hidden state

$$\mathbf{h}_j = f_\theta(\mathbf{h}_{j-1}, \texttt{<pause>}),$$

while suppressing any semantic output. A sequence of $K$ pause tokens thus yields $K$ steps of latent computation over the current cognitive state. We train a set of $K$ pause tokens (where $K$ exactly equals the number of generated latent tokens by memory weaver) under the same protocol in Equation (10).

**Results.** We report the ablation results as follows:

Table 9: The performance comparison between `MemGen` and a variant by replacing memory weaver with learnt pause tokens on three benchmarks. The backbone is set as `Qwen2.5-1.5B-Instruct`.

| Method | Kodcode | TriviaQA | GPQA |
|---|---|---|---|
| Trigger + Pause Token | 49.50 | 56.30 | 13.80 |
| Trigger + Weaver | **58.16** | **65.02** | **18.28** |

As shown in Table 9, although pause tokens provide moderate gains, indicating that allowing the model brief intervals of internal processing is beneficial, they consistently underperform the full weaver-equipped system. We attribute this to the fixed, context-independent nature of pause tokens: they cannot reconstruct or integrate task-relevant information in a targeted manner. In contrast, the memory weaver produces *context-sensitive* latent memory vectors tailored to the reasoner's current cognitive state, leading to substantially stronger performance.

## F INTEGRATION WITH RETRIEVAL-BASED MEMORY

### F.1 FORMALIZING THE INTEGRATION PROCESS

While the primary mechanism of `MemGen` leverages the parametric knowledge encapsulated within the memory weaver $\mathcal{W}_{\text{weaver}}$, the framework is designed to be extensible, allowing for seamless integration with external, retrieval-based memory systems. This hybrid approach enables the weaver to synthesize latent memories that are informed by both its internalized experiences and a vast corpus of external information, thereby providing a richer and more comprehensive context to the reasoner $\pi_\theta$. Let $\mathcal{M}_{\text{ext}}$ denote an external memory database, and let $\mathcal{R}(\cdot)$ be a retrieval function that, given a natural language query, returns a set of relevant textual memory snippets. When the memory trigger $\mathcal{T}_{\text{trigger}}$ determines an invocation is necessary at the token-generation step $j$ (*i.e.*, $d_j = \texttt{INVOKE}$), the natural language text generated thus far serves as the query for the external memory system. This query, denoted as $q_{t,j}$, is produced by decoding the sequence of tokens generated up to that point:

$$q_{t,j} = \text{Decode}(\mathbf{z}_{t,<j}). \tag{21}$$

The retrieval process is then formalized as:

$$\mathcal{C}_t = \mathcal{R}(q_{t,j}; \mathcal{M}_{\text{ext}}), \tag{22}$$

where $\mathcal{C}_t = \{c_1, c_2, \ldots, c_P\}$ is a set of $P$ retrieved textual snippets. These snippets are subsequently encoded into a sequence of embeddings, $\mathbf{E}_t \in \mathbb{R}^{L_c \times d_{\text{model}}}$, where $L_c$ is the total length of the encoded text. This allows the weaver to process the retrieved information in its native latent space.

This retrieved information is subsequently merged with the reasoner's internal cognitive state $\mathbf{H}_{t,<j}$. The combined context is then fed into the memory weaver $\mathcal{W}_{\text{weaver}}$ to produce the final latent memory. The invocation of the weaver, as described in Equation (5), is thus modified to:

$$\mathbf{M}_t = \mathcal{W}_{\text{weaver}}([\mathbf{H}_{t,<j}; \mathbf{E}_t]), \tag{23}$$

where $[\cdot ; \cdot]$ denotes the concatenation of the hidden state sequences. This integrated process allows $\mathcal{W}_{\text{weaver}}$ to reconstruct both internal parametric knowledge and externally retrieved information into a compact, potent latent memory $\mathbf{M}_t$ for the reasoner.

## F.2 EXPERIMENTAL RESULTS

As shown in Table 10, even when `MemGen`'s own parametric memory is disabled (which means that merely the retrieved textual snippets are fed into $\mathcal{W}_{\text{weaver}}$), `MemGen` significantly enhances the retrieval baseline, boosting performance on ALFWorld from $36.18\%$ to $45.60\%$ and on PopQA from $28.16\%$ to $39.50\%$. This demonstrates that `MemGen` serves as a powerful synthesizer, not merely appending but actively reconstructing externally retrieved information into a more potent, task-relevant latent memory, thereby underscoring the necessity of a generative memory mechanism. Most notably, when the weaver's internal parametric memory is enabled alongside external retrieval, performance is dramatically amplified, achieving scores of $75.90\%$ on ALFWorld and $76.40\%$ on TriviaQA. This highlights the powerful synergy achieved when the memory weaver is empowered to dynamically integrate its own internalized experiences with external textual knowledge.

Table 10: Performance results when integrating `MemGen` with a retrieval-based memory baseline, ExpeL. The backbone LLM is set as `SmolLM3-3B`.

| Method | ALFWorld | TriviaQA | PopQA |
|---|---|---|---|
| Vanilla LLM | 18.96 | 10.47 | 8.23 |
| ExpeL | 36.18 | 46.20 | 28.16 |
| `MemGen` + ExpeL (*w/o* parametric memory) | 45.60 | 53.20 | 39.50 |
| `MemGen` + ExpeL (*w/* parametric memory) | **75.90** | **76.40** | **60.23** |

## G   LATENT MEMORY TOKEN DEMONSTRATION

---

**Cases from TriviaQA**

**Case 1**
**system prompt**
Answer the given question. You must conduct reasoning inside <think> and </think> first
    every time you get new information. After reasoning, if you find you lack some
    knowledge, you can call a search engine by <search> query </search> and it will
    return the top searched results between <information> and </information>. You can
    search as many times as your want. If you find no further external knowledge
    needed, you can directly provide the answer inside <answer> and </answer>, without
    detailed illustrations. For example, <answer> Beijing </answer>.

**user prompt**
Which Lloyd Webber musical premiered in the US on 10th December 1993?

**assistant output**
**[UPPORT...',enable certif]**<think> I need to find out which Lloyd Webber musical
    premiered in the US on 10th December 1993. </think>
**[essengeryyyyMMddELCOME certif]**<search> Lloyd Webber musical premiered in the US on 10
    th December 1993 </search>

**user prompt**
<observation> Doc 1(Title: "Andrew Lloyd Webber") opened at the Adelphi Theatre in
    London on 12 July 1993, and ran for 1,529 performances ... </observation>

**assistant output**
**[essengeryyyyMMddELCOME
    certif]**<think> The search results indicate that "Sunset Boulevard" is the Lloyd
    Webber musical that premiered in the US on 10th December 1993. </think>
**[UPPORT...',enable certif]**<answer> Sunset Boulevard </answer>

**Case 2**
**system prompt**
Answer the given question. You must conduct reasoning inside <think> and </think> first
    every time you get new information. After reasoning, if you find you lack some
    knowledge, you can call a search engine by <search> query </search> and it will
    return the top searched results between <information> and </information>. You can
    search as many times as your want. If you find no further external knowledge

---

```
            needed, you can directly provide the answer inside <answer> and </answer>, without
             detailed illustrations. For example, <answer> Beijing </answer>.

user prompt
Who had a 70s No 1 hit with Kiss You All Over?

assistant output
[UPPORT...',eniable certif]<think> I need to search who had a 70s No 1 hit with the
      song "Kiss You All Over." </think>
[essengeryyyyMMddELCOME
      certif]<search> who had a 70s No 1 hit with Kiss You All Over? </search>

user prompt
<observation> Doc 1(Title: "Kiss You All Over") Exile's only big hit in the pop rock
      market (they would later have great success on the country music charts)... </
      observation>

assistant output
[essengerUPPORT'options certif]<think> The song "Kiss You All Over" was a 70s No 1 hit
      with the band Exile. </think>
[UPPORT...',eniable certif]<answer> Exile </answer>
```

**Case 3**
**system prompt**
```
Answer the given question. You must conduct reasoning inside <think> and </think> first
       every time you get new information. After reasoning, if you find you lack some
      knowledge, you can call a search engine by <search> query </search> and it will
      return the top searched results between <information> and </information>. You can
      search as many times as your want. If you find no further external knowledge
      needed, you can directly provide the answer inside <answer> and </answer>, without
       detailed illustrations. For example, <answer> Beijing </answer>.
```

**user prompt**
```
What was the name of Michael Jackson's autobiography written in 1988?
```

**assistant output**
```
[UPPORTUPPORTeniable certif]<think> I need to search for the name of Michael Jackson's
      autobiography written in 1988. </think>
[UPPORT...',eniable certif]<search> Michael Jackson autobiography 1988 </search>
```

**user prompt**
```
<observation> Doc 1(Title: "Michael Jackson") of 4.4 million people. In 1988, Jackson
      released his only autobiography, ""Moonwalk"", which took four years to complete
      and sold 200,000 copies... </observation>
```

**assistant output**
```
[cmsUPPORT'options Certif]<think> The name of Michael Jackson's autobiography written
      in 1988 is "Moonwalk". </think>
[UPPORTUPPORTeniable certif]<answer> Moonwalk </answer>
```

**Case 4**
**system prompt**
```
Answer the given question. You must conduct reasoning inside <think> and </think> first
       every time you get new information. After reasoning, if you find you lack some
      knowledge, you can call a search engine by <search> query </search> and it will
      return the top searched results between <information> and </information>. You can
      search as many times as your want. If you find no further external knowledge
      needed, you can directly provide the answer inside <answer> and </answer>, without
       detailed illustrations. For example, <answer> Beijing </answer>.
```

**user prompt**
```
Which volcano in Tanzania is the highest mountain in Africa?
```

**assistant output**
```
[UPPORT...',eniable certif]<think> I need to search for the highest mountain in Africa,
       which is located in Tanzania. </think>
[UPPORTUPPORTeniable certif]<search> highest mountain in Africa in Tanzania </search>
```

**user prompt**
```
<observation> Doc 1(Title: "Mount Kilimanjaro") Mount Kilimanjaro Mount Kilimanjaro or
      just Kilimanjaro (), with its three volcanic cones, ""Kibo"", ""Mawenzi"", and ""
      Shira"", is a dormant volcano in Tanzania... </observation>
```

**assistant output**
```
[essengeryyyyMMddeniable certif]<think> The highest mountain in Africa located in
      Tanzania is Mount Kilimanjaro. </think>
[UPPORT...',eniable certif]<answer> Mount Kilimanjaro </answer>
```

---

**Cases from KodCode**

**Case 1**
**MODEL INPUT:**
Write an efficient and correct Python function to solve the following problem.Question:
    Write a function that takes a list of integers and returns a new list containing
    only the even integers from the original list, sorted in non-decreasing order. The
    list can contain negative numbers and is not necessarily sorted. If the input
    list is empty or contains no even integers, the function should return an empty
    list.

**MODEL OUTPUT:**
**[keyword-kindërgetAs-slide]**def even_sorted(lst):
**[.keyword_pick]** """
**[LTRetical]** Returns a new list containing only the even integers from the original list
    ,
    sorted in non-decreasing order.
**[.keyword_pick]** """
**[LTRetical]** even_numbers = [num for num in lst if num % 2 == 0]
**[.keyword_pick]** even_numbers.sort()
**[.keywordetical]** return even_numbers<|im_end|>

**Case 2**
**MODEL INPUT:**
Write an efficient and correct Python function to solve the following problem.Question:
    Write a python function to find the first maximum and the second maximum number
    in an array. If the array has less than two unique elements, return 'None' for the
    missing values.

**MODEL OUTPUT:**
**[keyword-kindërgetAs-slide]**def find_first_max_second_max(arr):
    """
**[.keywordetical]** Returns the first and second maximum numbers in the array.
    If the array has less than two unique elements,**[.keyword_pick]** returns None for the
        missing values.
**[.keyword_pick]** """
    if len(arr) < 2:
**[.keyword_pick]** return None,**[.keyword_pick]** None

**[.keywordetical]** unique_elements = set(arr)
**[.keyword_pick]** if len(unique_elements) < 2:
**[.keywordetical]** return None,**[.keyword_pick]** None

**[.keywordetical]** first_max = max(unique_elements)
    second_max = max(unique_elements, default=None)

    return first_max, second_max<|im_end|>

**Case 3**
**MODEL INPUT:**
Write an efficient and correct Python function to solve the following problem.Question:
    Given a list of integers, write a function that returns the length of the longest
    contiguous subarray where all elements have the same value. The function should
    also return the value of the elements in this subarray. Consider the output to be
    a tuple where the first element is the length of the longest subarray and the
    second element is the value of the elements in the subarray.

For example, consider the following list as input,
numbers = [1, 2, 2, 3, 3, 3, 2, 2, 2, 2, 1, 1]

Expected Output: (4, 2)

**MODEL OUTPUT:**
**[kindgetAs?-kindërgetAs-slide]**def longest_subarray_with_same_value(numbers):
    """
**[.keyword_pick]** Returns the length of the longest contiguous subarray where all
    elements have the same value.
**[.keyword_pick]** The function also returns the value of the elements in this subarray.
**[LTR_pick]** """
**[LTRetical]** if not numbers:
**[.keyword_pick]** return (0, None)

    max_length = 1
**[.keyword_pick]** current_length = 1
    max_value = numbers[0]
    current_value = numbers[0]

**[.keyword_pick]** for i in range(1,**[LTR_pick]** len(numbers)):
**[LTR_pick]** if numbers[i] == current_value:

```
        current_length += 1
      else:
          if current_length > max_length:
[LTR_pick] max_length = current_length
            max_value = current_value
          current_length = 1
          current_value = numbers[i]

   if current_length > max_length:
      max_length = current_length
      max_value = current_value

   return (max_length, max_value)<|im_end|>
```

## Cases from GSM8K

**Case 1**
**MODEL INPUT:**
Solve the math problem with proper reasoning, and make sure to put the FINAL ANSWER
    inside \boxed{}.
Question: Billy sells DVDs. He has 8 customers on Tuesday. His first 3 customers buy
    one DVD each. His next 2 customers buy 2 DVDs each. His last 3 customers don't buy
    any DVDs. How many DVDs did Billy sell on Tuesday?

**MODEL OUTPUT:**
**[ even.  s.  even even ...**
    **even]** Billy's first 3 customers buy 3 * 1 = <<3*1=3>>3 DVDs.
**[-gallery.]**His next 2 customers buy 2 * 2 = <<2*2=4>>4 DVDs.
**[-routing.]**Billy sells 3 + 4 = <<3+4=7>>7 DVDs on Tuesday.\boxed{7}<|im_end|>

**Case 2**
**MODEL INPUT:**
Solve the math problem with proper reasoning, and make sure to put the FINAL ANSWER
    inside \boxed{}.
Question: Henry made two stops during his 60-mile bike trip. He first stopped after 20
    miles. His second stop was 15 miles before the end of the trip. How many miles did
     he travel between his first and second stops?

**MODEL OUTPUT:**
**[.....  a even even even...  even**
    **]**Henry traveled 60-15=<<60-15=45>>45 miles before his second stop.
**[-android.]**He traveled 20 miles before his first stop,**[ecies.]** so he traveled
    45-20=<<45-20=25>>25 miles between his first and second stops.\boxed{25}<|im_end|>
**[ even.]**The final answer is 25<|im_end|>

**Case 3**
**MODEL INPUT:**
Solve the math problem with proper reasoning, and make sure to put the FINAL ANSWER
    inside \boxed{}.Question: A basket contains 25 oranges among which 1 is bad, 20%
    are unripe, 2 are sour and the rest are good. How many oranges are good?

**MODEL OUTPUT:**
**[. even..  pre.  even even even...  even**
    **]**The number of unripe oranges is 20/100*25 = <<20/100*25=5>>5
**[-animate.]**The number of sour oranges is 2
**[-login.]**The number of bad oranges is 1
**[-login.]**The number of good oranges is 25 - 5 - 2 - 1 = <<25-5-2-1=17>>17\boxed{17}<|
    im_end|>

## Cases from GSM8K Cluster

**Cluster 1**
Solve the math problem with proper reasoning, and make sure to put the FINAL ANSWER
    inside \boxed{}.Question: Josh decides to try flipping a house. He buys a house
    for $80,000 and then puts in $50,000 in repairs. This increased the value of the
    house by 150%. How much profit did he make?
**[.....  pre.  even even even...  even ]**The house was worth $80

Solve the math problem with proper reasoning, and make sure to put the FINAL ANSWER
    inside \boxed{}.Question: James decides to run 3 sprints 3 times a week. He runs
    60 meters each sprint. How many total meters does he run a week?
**[.....  out.  even even even..  even even ]**He runs 3*60=<<3*60=180>>180 meters a day

```
Solve the math problem with proper reasoning, and make sure to put the FINAL ANSWER
     inside \boxed{}.Question: Kylar went to the store to buy glasses for his new
     apartment. One glass costs $5, but every second glass costs only 60% of the price.
      Kylar wants to buy 16 glasses. How much does he need to pay for them?
[.... pre... even... even ]The first 16 glasses cost 16 * 5 = $<<16*5=80>>80

Solve the math problem with proper reasoning, and make sure to put the FINAL ANSWER
     inside \boxed{}.Question: Toulouse has twice as many sheep as Charleston.
     Charleston has 4 times as many sheep as Seattle. How many sheep do Toulouse,
     Charleston, and Seattle have together if Seattle has 20 sheep?
[. even even.  s.  even even even..  even even
     ]Charleston has 4*20=<<4*20=80>>80 sheep.

Solve the math problem with proper reasoning, and make sure to put the FINAL ANSWER
     inside \boxed{}.Question: John drives for 3 hours at a speed of 60 mph and then
     turns around because he realizes he forgot something very important at home. He
     tries to get home in 4 hours but spends the first 2 hours in standstill traffic.
     He spends the next half-hour driving at a speed of 30mph, before being able to
     drive the remaining time of the 4 hours going at 80 mph. How far is he from home
     at the end of those 4 hours?
[...... even even even... even
     ]He drives 60*3 = <<60*3=180>>180 miles in the first 3 hours.

Cluster 2
Janet's ducks lay 16 eggs per day,[-animate.] and she eats 3 eggs for breakfast every
     morning

Wendi feeds her 20 chickens 15 cups of feed in the morning,[-animate.] 25 cups in the
     afternoon

She sold half of what was left at the orange house,[-animate.] she had 1/2 left.

x = 162/18 = <<162/18=9>>9
[-animate.]Allen's age is 11x = 11*9 = <<11*9=99>>99 years old now.

He runs 20 miles a day because 60/3=<<60/3=20>>20
[-animate.]He runs 10 miles the other two days because 20/2=<<20/2=10>>10

Cluster 3
It takes 2/2=<<2/2=1>>1 bolt of white fiber.
[ecies.]So it takes 2+1=<<2+1=3>>3 bolts in total.

The total cost of the five cell phones is $150 x 5 = $<<150*5=750>>750.
[ even.]The interest for each unit is $150 x 2/100 = $<<150*2/100=3>>3.

The second level is 16*2=<<16*2=32>>32 square footage.
[ even.]The third level is 32*2=<<32*2=64>>64 square footage.

20% of 20 students is 20 x 20/100 = <<20*20/100=4>>4 students.
[ecies.]The remaining students are 20 - 4 = <<20-4=16>>16 students.

If Artie rounds the price of petunias to the nearest dollar,[-animateecies] he will
     sell them for $2 each.

Cluster 4
The groomer offers a 30% discount,[-loginecies] so the cost of the grooming is 100 -
     (100 * 0.[ecies.]30) = $<<100-30=70>>70

If Dana skips at 3 miles per hour,[-login.] then she runs at 3 * 4 = <<3*4=12>>12 miles
      per hour.

There were 140 * 2 = <<140*2=280>>280 emeralds in the chest.
[-login.]There were 175 + 140 + 280 = <<175+140+280=595>>595 gems in the chest.

There are 30+60 = <<30+60=90>>90 students at Dr.[-loginecies.] Wertz's school.

Cecilia will feed her puppy 1 cup of dog food every day for 180 days,[-login.] so she
     will use 180 cups of dog food in the first 180 days.
```

# H  MEMORY FUNCTIONAL STUDY

## H.1  VISUALIZATION PROCESS

This section details the methodology employed to visualize the geometric structure of the latent memory (as shown in Figures 5 and 11) and to identify semantically distinct clusters within it. The

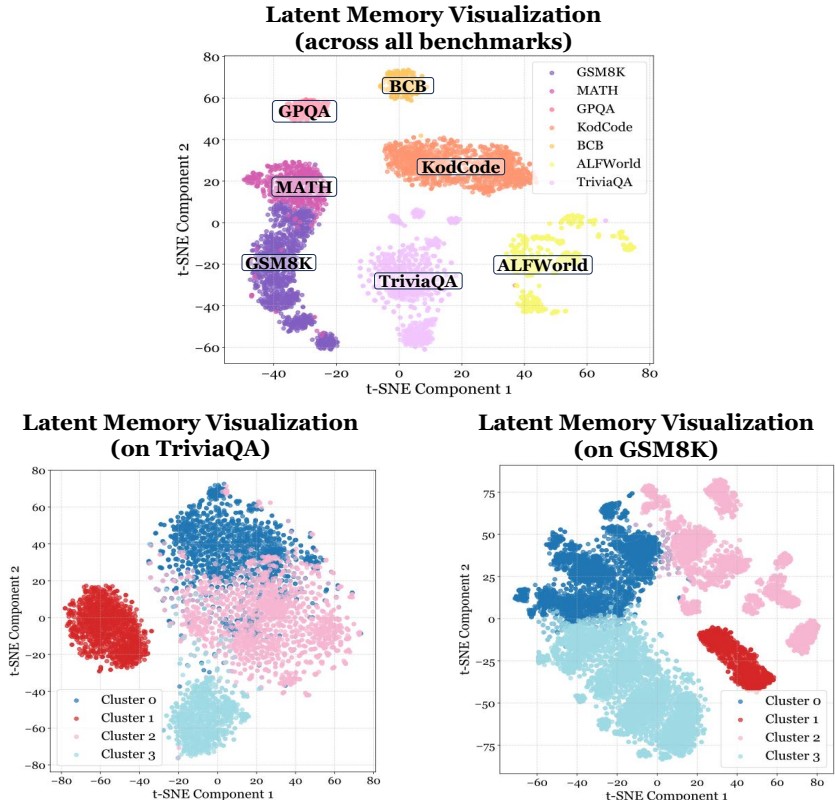

Figure 11: (*Up*) t-SNE visualization of latent memories generated by `MemGen` +`SmolLM3-3B` across datasets; (*Down*) Latent memory visualization within the TriviaQA and GSM8K datasets, clustered using $K$-means.

process involves two primary stages: obtaining a fixed-dimensional representation for each memory sequence and applying dimensionality reduction and clustering algorithms.

**Sequence Representation.** Let $\mathcal{D}_{\mathrm{mem}} = \{\mathbf{M}_1, \mathbf{M}_2, \ldots, \mathbf{M}_Q\}$ be a corpus of $Q$ latent memory sequences collected from `MemGen`'s inference, where each sequence $\mathbf{M}_i = (\mathbf{m}_{i,1}, \ldots, \mathbf{m}_{i,K}) \in \mathbb{R}^{K \times d_{\mathrm{model}}}$ consists of $K$ token embeddings of dimension $d_{\mathrm{model}}$. To obtain a single, holistic vector representation for each sequence, we compute its mean embedding $\bar{\mathbf{m}}_i \in \mathbb{R}^{d_{\mathrm{model}}}$:

$$\bar{\mathbf{m}}_i = \frac{1}{K} \sum_{l=1}^{K} \mathbf{m}_{i,l}. \tag{24}$$

This procedure yields a set of high-dimensional representations $\bar{\mathcal{M}} = \{\bar{\mathbf{m}}_1, \bar{\mathbf{m}}_2, \ldots, \bar{\mathbf{m}}_Q\}$, which serves as the input for the subsequent analysis.

For visualization, we employed t-SNE (Maaten & Hinton, 2008) to project the high-dimensional set $\bar{\mathcal{M}}$ into a two-dimensional space. Formally, this mapping can be expressed as $\mathbf{y}_i \in \mathbb{R}^2 = f_{\text{t-SNE}}(\bar{\mathbf{m}}_i)$. These 2D data points are then utilized for the scatter plot.

To quantitatively identify distinct functional groups, we applied the K-means algorithm directly to the high-dimensional mean embeddings in $\bar{\mathcal{M}}$. This procedure partitions the memory representations into a predefined number of $N$ discrete clusters, $\mathcal{C} = \{C_1, C_2, \ldots, C_N\}$ (we set $N = 4$).

### H.2 FAILURE TAXONOMY DEFINITIONS

To systematically analyze the functional impact of different memory clusters, we adopt and refine the failure taxonomy proposed by Song et al. (2025). Below, we provide precise definitions of

each failure category in the context of LLM-based agent behavior, which guide our annotation and evaluation process:

- **Planning Failure.** This category refers to errors in high-level task decomposition and strategic decision-making. The agent either formulates subgoals that do not align with the original objective, fails to plan intermediate steps necessary for task completion, or misorders the reasoning sequence, leading to suboptimal or incomplete solutions.

- **Compositional Reasoning.** This type of failure occurs when the agent struggles to integrate multiple pieces of information or reasoning steps into a coherent solution.

- **Tool Parsing Error.** This failure occurs when the agent's generated output cannot be parsed into a valid tool call. Typical patterns include malformed function signatures, missing required arguments, or unrequired arguments in the invocation, which prevent the external tool from being executed as intended.

- **Tool Response Error.** This category refers to errors that arise after a tool has been successfully invoked. Either the tool itself returns incorrect or incomplete information, or the agent misuses or misinterprets the returned content.

- **Answer Formatting Failure.** This category includes errors in the final presentation or structuring of the output, despite the reasoning process being largely correct. Examples include incorrect output format (*e.g.*, unable to include the answers with in \boxed), violation of task-specific response templates, or missing required components in the final answer.

- **Demand Misunderstanding.** This failure indicates that the agent has misinterpreted the user's intent or the task specification. It may solve a different problem from the one posed, ignore key constraints, or pursue irrelevant objectives due to misunderstanding the instruction semantics.

- **Think-Act Inconsistency.** This type refers to a mismatch between the agent's internal reasoning and its subsequent action in ReAct mode. The agent may articulate a correct reasoning chain but execute a contradictory action or produce a final answer inconsistent with its prior deliberation.

- **False Belief** captures cases where the agent maintains and reasons with inaccurate assumptions about the external environment, user state, or task context.

### H.3 ANNOTATING FAILURE MODES AND FILTERING LATENT MEMORY

**Annotation of Failure Modes.** We manually annotated agent failures on the TriviaQA dataset based on the eight failure modes discussed above. Each trajectory generated by the agent was evaluated by human annotators who assigned only one failure label if the agent's behavior deviated from a successful path. Trajectories exhibiting failures that did not fall into these predefined categories were excluded from this specific analysis.

**Inference-time Filtering of Latent Memory Clusters.** The core of our intervention study involved selectively removing the influence of a specific latent memory cluster during the agent's reasoning process. Our methodology is designed to be consistent with the clustering process itself, which operates on sequence-level representations.

Let $\mathcal{C} = \{C_1, C_2, \ldots, C_N\}$ be the set of $N$ latent memory clusters. These clusters were derived by applying $K$-means to a collection of historical memory representations, where each representation is the mean embedding of an entire latent memory sequence. For each cluster $C_i \in \mathcal{C}$, we compute its centroid $\mu_i \in \mathbb{R}^{d_{\text{model}}}$ by averaging these historical sequence representations within the cluster.

During inference, when the memory weaver synthesizes a new latent memory sequence $\mathbf{M}_t = (\mathbf{m}_{t,1}, \ldots, \mathbf{m}_{t,K}) \in \mathbb{R}^{K \times d_{\text{model}}}$, we first compute its single-vector representation, $\bar{\mathbf{m}}_{\text{new}}$, by averaging its constituent token embeddings:

$$\bar{\mathbf{m}}_{\text{new}} = \frac{1}{K} \sum_{l=1}^{K} \mathbf{m}_{t,l}. \tag{25}$$

To determine the semantic affiliation of this new sequence, we compare its mean embedding $\bar{\mathbf{m}}_{\text{new}}$ against a comprehensive reference set $\mathcal{E}_{\text{comp}} = \mathbf{E}_{\text{vocab}} \cup \{\mu_1, \ldots, \mu_N\}$, where $\mathbf{E}_{\text{vocab}} \in \mathbb{R}^{V \times d_{\text{model}}}$ is the LLM's vocabulary embedding matrix. We then identify the set of top-$k$ nearest neighbors to $\bar{\mathbf{m}}_{\text{new}}$ based on cosine similarity, denoted as $\mathcal{S}_k(\bar{\mathbf{m}}_{\text{new}})$. In our experiments, we set $k = 10$.

To ablate the influence of a target cluster $C_j$, the entire latent memory sequence $\mathbf{M}_t$ is filtered (i.e., discarded and not prepended to the reasoner's context) if the centroid of that cluster, $\mu_j$, is present within this top-$k$ set. Formally, $\mathbf{M}_t$ is filtered if:

$$\mu_j \in \mathcal{S}_k(\bar{\mathbf{m}}_{\text{new}}). \tag{26}$$

This sequence-level filtering allows us to precisely ablate the contribution of a specific learned memory function and observe its impact on agent behavior.

### H.4   COMPARISON BETWEEN LATENT TOKENS AND TRAJECTORIES

For each memory invocation at step $j$ and timestep $t$, we construct three sets of embeddings within the hidden representation space of the frozen reasoner $\pi_\theta$. First, we record the latent memory tokens generated by the weaver,

$$\mathbf{M}_t = [\mathbf{m}_{t,1}, \ldots, \mathbf{m}_{t,K}] \in \mathbb{R}^{K \times d_{\text{model}}}.$$

Second, we obtain the corresponding context hidden states that serve as the hook for memory generation,

$$\mathbf{H}_{t,<j} = (\mathbf{h}_{t,1}, \ldots, \mathbf{h}_{t,j-1}) \in \mathbb{R}^{(j-1) \times d_{\text{model}}},$$

collected directly from the forward pass of $\pi_\theta$ during online reasoning. Third, we take a past successful trajectory $\tau_{\text{prev}}$ from $\mathcal{H}$ and replay it under teacher forcing through $\pi_\theta$, yielding its hidden-state sequence

$$\tilde{\mathbf{H}}^{\text{traj}} = (\tilde{\mathbf{h}}_1, \ldots, \tilde{\mathbf{h}}_L) \in \mathbb{R}^{L \times d_{\text{model}}}.$$

To enable joint visualization, we compute the mean embedding of each set:

$$\bar{\mathbf{m}}_t = \frac{1}{K} \sum_{k=1}^{K} \mathbf{m}_{t,k}, \qquad \bar{\mathbf{h}}_t^{\text{ctx}} = \frac{1}{j-1} \sum_{k=1}^{j-1} \mathbf{h}_{t,k}, \qquad \bar{\mathbf{h}}^{\text{traj}} = \frac{1}{L} \sum_{\ell=1}^{L} \tilde{\mathbf{h}}_\ell.$$

Finally, we project the three averaged vectors

$$\{\bar{\mathbf{m}}_t, \ \bar{\mathbf{h}}_t^{\text{ctx}}, \ \bar{\mathbf{h}}^{\text{traj}}\}$$

from $\mathbb{R}^{d_{\text{model}}}$ into two dimensions (via t-SNE) for visualization. The resulting visualizations are shown in Figure 12.

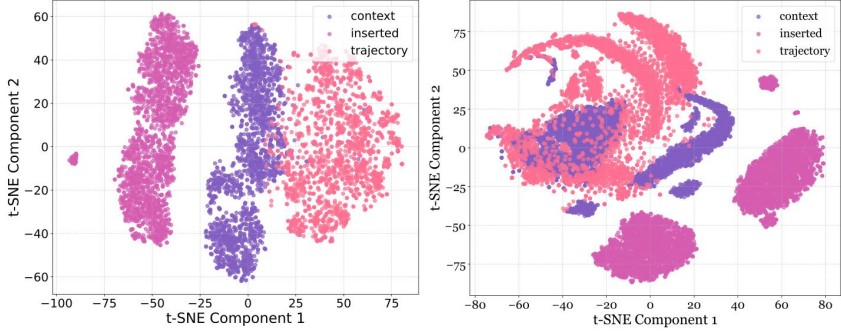

Figure 12: The t-SNE visualizations among the learnt latent tokens, trajectory representations and context representations. The left is drawn from KodCode dataset, and the right is from TriviaQA dataset.

As shown in Figure 12, the trajectory and context embeddings occupy more similar regions of the feature space, as both originate from transformations of human-readable tokens. In contrast, latent memory tokens form a more machine-native and human-unreadable representation, resulting in a substantially different embedding distribution.

We further quantify the geometric relationship between the three clusters by computing the Euclidean distances between their centers in $\mathbb{R}^{d_{\text{model}}}$. For each invocation $(t, j)$, we treat $\bar{\mathbf{m}}_t$, $\bar{\mathbf{h}}_t^{\text{ctx}}$, and $\bar{\mathbf{h}}^{\text{traj}}$ as the centers of the latent-memory, context, and trajectory clusters, respectively, and compute the pairwise distances

$$d_{\text{mem-ctx}}^{(t)} = \left\| \bar{\mathbf{m}}_t - \bar{\mathbf{h}}_t^{\text{ctx}} \right\|_2, \quad d_{\text{mem-traj}}^{(t)} = \left\| \bar{\mathbf{m}}_t - \bar{\mathbf{h}}^{\text{traj}} \right\|_2, \quad d_{\text{ctx-traj}}^{(t)} = \left\| \bar{\mathbf{h}}_t^{\text{ctx}} - \bar{\mathbf{h}}^{\text{traj}} \right\|_2.$$

Aggregating over all invocations $\mathcal{T}$, we report the mean distances

$$\bar{d}_{\text{mem-ctx}} = \frac{1}{|\mathcal{T}|} \sum_{t \in \mathcal{T}} d^{(t)}_{\text{mem-ctx}}, \quad \bar{d}_{\text{mem-traj}} = \frac{1}{|\mathcal{T}|} \sum_{t \in \mathcal{T}} d^{(t)}_{\text{mem-traj}}, \quad \bar{d}_{\text{ctx-traj}} = \frac{1}{|\mathcal{T}|} \sum_{t \in \mathcal{T}} d^{(t)}_{\text{ctx-traj}},$$

which directly reflect how far the latent memory centers deviate from the context and replayed-trajectory manifolds.

Table 11: Pairwise Euclidean distances between the centers of context, latent-memory (inserted), and trajectory clusters.

| Dataset | context $\leftrightarrow$ inserted | context $\leftrightarrow$ trajectory | inserted $\leftrightarrow$ trajectory |
|---------|-----------------------------------|--------------------------------------|---------------------------------------|
| GPQA | 165.311 | 64.180 | 185.315 |
| KodCode | 227.108 | 96.831 | 197.105 |

From Table 11, the distances reveal a consistent geometric pattern: the context and trajectory centers lie extremely close, while the generated latent memories occupy a distant region of the representation space, forming a well-separated cluster from both context and trajectory. This large separation demonstrates that the weaver does not merely compress or replay past hidden states; instead, it synthesizes novel latent structures that are parametrically reconstructed rather than retrieved, supporting our claim that latent memory introduces genuinely new inferential content beyond the observed trajectories.

