# OpenReview forum: "MemGen: Weaving Generative Latent Memory for Self-Evolving Agents"
_ICLR.cc/2026/Conference — ICLR 2026 Poster_

### Official Review · Reviewer_Z9t6 · 2025-10-28

**Soundness:** 3
**Presentation:** 2
**Contribution:** 4
**Rating:** 8
**Confidence:** 4

**Summary:**

This paper proposes MemGen, a latent generative memory framework designed to give LLM agents a more human-like and cognitively interwoven memory system. Unlike existing parametric memory (fine-tuning the base model and prone to catastrophic forgetting) or retrieval-based memory (externally stored context being rigid and extractive), MemGen introduces a two-component latent memory architecture: (1) a reinforcement-learned trigger that monitors the agent’s internal reasoning state and decides dynamically when memory should be invoked, and (2) a memory weaver that generates latent machine-native memory tokens on-the-fly rather than fetching or copying text. These latent memory tokens are interleaved back into the LLM’s hidden states during reasoning, enabling a recursive cycle of memory and thought without modifying the frozen backbone.

Experiments across several benchmarks show large performance gains. The authors further show MemGen spontaneously develops human-like memory faculties emerging without explicit supervision. They argue this demonstrates a pathway toward self-evolving agents with fluid cognitive memory, beyond prompt-stuffing or gradient-update-based methods.

The proposed approach is technically sound. MemGen operationalizes a genuine new memory paradigm by placing a reinforcement-learned trigger and a generative latent memory weaver directly into the LLM’s internal token-level reasoning loop, rather than relying on retrieval-append or parametric weight updates. This is architecturally meaningful and experimentally supported across domains, though some of the “human-like cognition” framing appears partly post-hoc. The paper is clearly structured and visually well supported, but occasionally heavy in rhetorical narration. Still, the overall contribution is high-value: MemGen establishes generative latent memory as a first-class alternative to retrieval-based or parametric memory, with a principled RL-trained trigger and real performance gains, making it an architecture-level contribution.

**Strengths:**

The paper introduces a genuinely new memory formulation based on generative latent memory, rather than yet another variation of retrieval concatenation or parametric weight-updating. The architecture is operational and not hypothetical. Memory is generated in latent token form and dynamically interwoven during reasoning at token-time, with a reinforcement-learned trigger that decides when memory should be invoked, rather than heuristically forcing memory at preset turn boundaries. This is a substantial architectural step. The evaluation is serious and shows not only higher in-domain performance, but also strong cross-domain generalization and non-trivial continual learning behavior. The authors go beyond accuracy metrics and perform targeted interventions on latent clusters to reverse-engineer memory roles, providing convincing evidence of emergent planning, procedural memory, and working memory patterns.

Another notable strength is the breadth and sincerity of experimentation, which I appreciate enormously. The authors evaluate across nine benchmarks and report non-incremental gains. MemGen outperforms both retrieval-based and RL-based agent memory baselines consistently, and unlike most memory systems, it demonstrates improvement even when evaluated out of training domain.

**Weaknesses:**

The paper leans too aggressively on cognitive analogies (for example, hippocampus-like reconstruction and human-like memory faculties) that are partially justified but still interpretive rather than causally established. The intervention studies are solid, but they still operate at the level of inferred function, not formal guarantees. Some readers will find the cognitive framing oversold relative to the evidence. The stronger claim is architectural novelty. The biological narrative should be toned down in messaging to match what is strictly demonstrated.

In terms of writing, the paper is dense to the point of being unnecessarily heavy, especially in the introduction and motivation. The authors repeat the retrieval-versus-parametric critique multiple times, instead of positioning MemGen directly and succinctly. There is also a conceptual risk of blurring with latent steering and latent CoT intervention work such as Medusa, CODI, Coconut, and related latent policy modulation methods. The differentiation is real, but it should be more sharply articulated to avoid the impression that this is simply another latent-control method rather than a true memory system. Clarity and restraint in framing would strengthen the paper further.

**Questions:**

(1) Your method operates in the latent space and applies interventions during generation. How do you conceptually and architecturally differentiate MemGen from existing latent intervention and latent CoT works such as CODI, Medusa, Coconut, Co-processor, SoftCoT? IOW what prevents this from being viewed as “latent policy steering” rather than specifically “latent memory”?

(2)  You claim that latent memory is reconstructed and not retrieved or embedded similarity-based. Can you provide quantitative evidence (not just intuition) that the weaver is actually generating novel inferential structure, not just embedding-compressed re-encodings of prior examples?

(3) Under what task conditions does MemGen approach the worst case of inference latency? If this grows with task horizon, can MemGen scale to multi-hour multi-hop agents realistically?

(4) Since the memory trigger is trained via RL, did you observe any pathological cases where it overfires or underfires in edge situations?

**Details Of Ethics Concerns:**

The work uses only academic benchmarks, does not involve sensitive data or human subjects, and is not deployed in real-world or safety-critical settings. I do not see any ethical concerns requiring escalation.

---

> ### Author Response · Authors · 2025-11-20
> **[Part 1/3] Author Response**
>
> We sincerely thank you for your careful comments and thorough understanding of our paper! Here we give point-by-point responses to your comments and describe the revisions we made to address them.
>
> ---
>
> > **`Weakness 1`** The paper leans too aggressively on cognitive analogies that are partially justified but still interpretive rather than causally established. The intervention studies are solid, but they still operate at the level of inferred function, not formal guarantees. .. The biological narrative should be toned down in messaging to match what is strictly demonstrated.
>
> Thank you for this valuable suggestion, which has greatly improved the clarity and readability of our manuscript! Following your suggestion, we have revised the text to de-emphasize biological analogies and better highlight the architectural contributions of our method. Several key changes are:
>
> - **Line 102**: we removed references to human-like memory architectures and instead emphasize the functional distinctions within MemGen.
> - **Beginning of Section 4.1**: we eliminated the metaphors involving the frontoparietal system and hippocampus.
> - **Section 2, Related Work**: We removed the emphasis on "human-esque" memory architecture.
> - **Above Eq. (6)**: we removed the comparison between the weaver and hippocampal consolidation.
>
> Additional refinements are also highlighted in the updated manuscript PDF.
>
> ---
>
> > **`Weakness 2`** In terms of writing, the paper is dense to the point of being unnecessarily heavy, especially in the introduction and motivation. The authors repeat the retrieval-versus-parametric critique multiple times, instead of positioning MemGen directly and succinctly.
>
> Thank you once again for your thoughtful feedback. Following your recommendation, we have revised the manuscript to reduce repeated retrieval-versus-parametric critiques and instead foreground the distinctive contributions of MemGen. Specifically:
>
> - **On Line 91**, we removed the phrase “moving beyond the static and extractive paradigm of prior memory systems” and now directly emphasize the core characteristics of MemGen.
> - **At the beginning of Section 4.1**, we eliminated redundant critiques of retrieval-based memory to focus on MemGen’s design.
> - **In Section 2 (Related Work)**, we removed "*rather than purely retrieval-based, nature*" and highlight the reasoning–memory interweaving that differentiates MemGen.
>
> Additional refinements are reflected in the updated manuscript PDF. We truly appreciate your insightful feedback, which has strengthened the rigor of the paper!
>
> ---
>
> > **`Weakness 3`** There is also a conceptual risk of blurring with latent steering and latent CoT intervention work. The differentiation is real, but it should be more sharply articulated to avoid the impression that this is simply another latent-control method rather than a true memory system.
>
>
> Thank you for the valuable suggestion! we agree that it is important to more clearly differentiate MemGen from prior work on *latent policy steering* and *latent CoT intervention*. To strengthen this distinction, we provide the following comparison table:
>
> | Method | How latent tokens are generated| Finetune backbone LLM? | Extra training parameters | Triggering mechanism| Insertion location|Multi-turn agent?| Continual learning? |Cross-task generalization?|
> |-|-|-|-|-| -| -| -| -|
> |Coconut| by reasoner | ✅  | ❌| – |every token|❌| ❌  | ❌ |
> |CODI | by reasoner  | ✅  | ❌ | –|every token| ❌ | ❌ | ❌|
> |Medusa  | by smaller LM  | ❌  | ✅ (decoding heads) | every token | every token | ❌  | ❌ | – |
> |Co-processor | by an LLM identical to the reasoner|❌| ✅ (as large as reasoner) | fixed  | fixed (decoding start)|❌|❌| ❌|
> |SoftCoT | by smaller LM | ❌ | ✅ (smaller LM)| fixed  | fixed (decoding start)  | ❌  | ❌ | ❌  |
> |**MemGen**| by weaver (LoRA) | ❌ | ✅ (two LoRAs) | dynamically judged by trigger | dynamic during decoding | ✅| ✅ | ✅ |
>
> Overall, we highlight three key aspects that most clearly distinguish MemGen from prior latent-steering and latent-CoT approaches:
>
> 1. **Dynamic interweaving with reasoning.** MemGen allows latent tokens to be inserted *dynamically* throughout decoding, a level of granularity and flexibility absent from prior methods. This tight interleaving of “reasoning” and “latent memory” is precisely why we conceptualize the system as a **latent memory** module.
> 2.  **Memory-oriented capabilities.** MemGen explicitly supports multi-turn agents, continual learning, and cross-task generalization, which capabilities we believe a true memory system should possess.
> 3. **Lightweight design.** Unlike Coconut, which requires full finetuning of the backbone LLM, or SoftCoT, which requires training an additional small LM, MemGen uses only two lightweight LoRA modules, making it highly parameter-efficient.
>
>
> Thank you once again for the insightful suggestion. We sincerely hope this clarification helps better situate MemGen’s unique position and contributions.

---

> ### Author Response · Authors · 2025-11-20
> **[Part 2/3] Author Response**
>
> > **`Question 1`** Your method operates in the latent space and applies interventions during generation. How do you conceptually and architecturally differentiate MemGen from existing latent intervention and latent CoT works such as CODI, Medusa, Coconut, Co-processor, SoftCoT? IOW what prevents this from being viewed as “latent policy steering” rather than specifically “latent memory”?
>
> Thank you once again! In our response to **`Weakness 3`**, we clarified along several dimensions why MemGen should be understood as a form of latent memory.
>
>
> ---
> > **`Question 2`** You claim that latent memory is reconstructed and not retrieved or embedded similarity-based. Can you provide quantitative evidence (not just intuition) that the weaver is actually generating novel inferential structure, not just embedding-compressed re-encodings of prior examples?
>
> To address your insightful question, we analyzed the learned latent memory tokens in comparison with past trajectories and the current context, both visually and quantitatively, to demonstrate that latent memory tokens are not mere restatements of prior examples or the present context. Instead, they introduce genuinely new information.
>
> Concretely, we construct three embedding clusters:
> (i) the **latent memory tokens** synthesized by the weaver,
> (ii) the **context hidden states** $\mathbf{H}\_{t,<j}$ that triggered the memory invocation, and
> (iii) the **hidden states of a past successful trajectory**, obtained by replaying that trajectory under teacher forcing.
>
> **Visualization** We first visualize the three clusters (details added in Appendix H.4). The context and trajectory embeddings appear noticeably closer in the t-SNE space, likely because both derive from transformations of human-readable tokens. In contrast, the latent memory tokens form a separate cluster located much farther away, reflecting their machine-native and human-unreadable nature.
>
> **Quantitative analysis** To further quantify these relationships, we compute the Euclidean distances between the centers of the three embedding clusters. As shown in the updated results, the context–trajectory distance remains consistently small across datasets, whereas the latent-memory cluster is an order of magnitude more distant from both. This substantial separation indicates that the weaver does not simply compress or replay existing hidden states. Rather, it synthesizes novel latent representations that are structurally distinct from both the current context and previously observed trajectories.
>
>
>
> | Dataset | context ↔ latent memory | context ↔ trajectory | latent memory ↔ trajectory |
> |-|-|-|-|
> | GPQA    | 165.311 | 64.180 | 185.315 |
> | KodCode | 227.108  | 96.831 | 197.105|

---

> ### Author Response · Authors · 2025-11-20
> **[Part 3/3] Author Response**
>
> > **`Question 3`** Under what task conditions does MemGen approach the worst case of inference latency? If this grows with task horizon, can MemGen scale to multi-hour multi-hop agents realistically?
>
>
> Thank you very much for this insightful question! To address it, we first provide a complexity analysis of MemGen’s inference latency, and then complement this with empirical evidence.
>
>
> **Latency Complexity Analysis.** Let $L$ be the number of tokens to generate, $C\_0$ the initial context length, $K$ the fixed latent memory length, and $d\_j \in \\{0, 1\\}$ the binary invocation decision at step $j$. For modern transformer architectures with KV caching, the latency of a single forward pass is approximately linear in the sequence length. We denote these latencies by $\mathcal{L}\_{\text{fwd}}(n) \approx c \cdot n$ for the base reasoner, and $\mathcal{L}\_{\text{trigger}}(n) \approx c\_T \cdot n$ and $\mathcal{L}\_{\text{weaver}}(n) \approx c\_W \cdot n$ for the lightweight trigger and weaver adapters, with hardware constants $c\_T, c\_W \ll c$.
>
> The latency of a vanilla model is
> $$
> T_{\text{vanilla}}
> = \sum_{j=1}^{L} \mathcal{L}_{\text{fwd}}(C_0 + j - 1)
> \approx \frac{c}{2} L^2.
> $$
>
> MemGen introduces an overhead $\Delta T$ composed of direct and indirect costs,
> $$
> \Delta T = \Delta T_{\text{direct}} + \Delta T_{\text{indirect}}.
> $$
>
> The direct cost is
> $$
> \Delta T\_{\text{direct}}= \sum\_{j=1}^{L}(\mathbb{I}[z\_{j-1} \in \mathcal{D}] \cdot \mathcal{L}\_{\text{trigger}}(C\_j)+ d_j \cdot \mathcal{L}\_{\text{weaver}}(C\_j)),
> $$
> and the dominant indirect cost is
> $$
> \Delta T\_{\text{indirect}}= \sum\_{j=1}^{L}\Big(\mathcal{L}\_{\text{fwd}}(C\_j)- \mathcal{L}\_{\text{fwd}}(C\_0 + j - 1)\Big),
> $$
> where $C\_j = C_0 + j - 1 + K \sum\_{k=1}^{j-1} d_k.$ Let $\bar{p} = (\sum\_{j=1}^{L} d\_j)/L$ be the average invocation rate (encouraged to be small via trigger RL), and let $\delta$ be the fraction of tokens that are delimiters. For large $L$, the overheads can be approximated as
> $$
> \Delta T\_{\text{indirect}} \approx \frac{c K \bar{p}}{2} L^2,
> \qquad
> \Delta T\_{\text{direct}} \approx \frac{\delta c\_T + \bar{p} c\_W}{2} L^2.
> $$
>
> Hence, the total overhead becomes
> $$
> \Delta T \approx \frac{1}{2}
> \bigl(c K \bar{p} + \delta c_T + \bar{p} c\_W\bigr) L^2.
> $$
>
> To evaluate how this overhead scales relative to the vanilla baseline as $L$ increases, we consider the asymptotic ratio:
> $$
> \lim\_{L \to \infty} \frac{\Delta T}{T\_{\text{vanilla}}}
> \approx
> K \bar{p}
> +
> \frac{\delta c\_T + \bar{p} c\_W}{c}.
> $$
>
> This shows that the *relative* overhead does not grow with the task horizon. Instead, it converges to a constant determined by the fixed hyperparameters $(K, c, c_T, c_W)$ and the learned invocation statistics $(\bar{p}, \delta)$. As $L$ becomes large in multi-hop agents, the total latency naturally increases, but the fraction attributable to MemGen remains fixed and small. Because the trigger is explicitly trained for sparse activation, $\bar{p}$ remains low, ensuring modest overhead in practice.
>
>
> **Empirical Evidence.** Finally, we note that Table 7 of our manuscript already provides an efficiency analysis of MemGen in a multi-turn agent setting. On ALFWorld, which typically requires 10–30 environment interaction steps, MemGen increases the inference latency of Qwen3-8B by only 2.63% relative to the SFT baseline, while yielding improved performance. We view this as empirical support that MemGen can be extended to more complex, long-horizon tasks with manageable overhead.
>
> Combining the theoretical complexity analysis with these empirical results, we respectfully argue that MemGen remains practical even as the task horizon grows and can, in principle, scale to substantially longer multi-hop scenarios.
>
>
>
>
>
> ---
>
>
> > **`Question 4`** Since the memory trigger is trained via RL, did you observe any pathological cases where it overfires or underfires in edge situations?
>
>
>
> Thank you for this valuable question. We address your concerns point by point:
>
> - **Excessive trigger activation**
> Recall from Eq. (8) that we include an explicit penalty on the number of trigger activations. This penalty is designed precisely to discourage overly frequent invocation. Empirically, we do not observe excessive activation during evaluation, which we attribute to the effectiveness of this regularization term.
> - **Insufficient trigger activation**
> As shown in Figures 4 and 9 of the manuscript, a trigger trained on one dataset (e.g., GSM8K) learns to activate more often within its own domain, but activates less frequently in moderately related domains (e.g., GPQA), and even less in distant domains (e.g., coding tasks such as KodCode). Empirically, this suggests that the trigger exhibits a form of domain-aware behavior: the farther the target task deviates from its training domain, the fewer activations it produces.
>
> Thank you once again for your careful and constructive review!

---

### Official Review · Reviewer_iSzn · 2025-10-30

**Soundness:** 4
**Presentation:** 3
**Contribution:** 3
**Rating:** 8
**Confidence:** 4

**Summary:**

This paper introduces a novel framework to dynamically incorporate memory during LLM reasoning. It uses a 'memory trigger' to determine when to use memory and a 'memory waver' to generate the memory to use. Through this design, they can achieve higher flexibility and capacity compared to the previous parametric memory and retrieve-based memory. Compared to other latent memory methods, their method has 'seamless interleaving of reasoning and memory', and can 'generatively reconstruct memory into novel, coherent insights' instead of using memory in a retrieval-based method.

**Strengths:**

The paper is well structured in general, and most of its explanations and descriptions are clear and intuitive. The proposed framework is pretty interesting and promising. Moreover, they provide extensive experiments that are relatively comprehensive and persuasive.

**Weaknesses:**

1. Although the paper is generally well written and clear, some parts are still somewhat confusing and insufficiently explained.
    * Sections 4.2 and 4.3 seemed confusing to me before reading Appendix C and D. Both 'memory trigger' and 'memory waver' require an instance of each other for their own training. This may confuse readers, and is only explained in Appendix D. Considering that Appendix D is not referenced in the main text until the end of Section 5.1, readers may find it difficult to identify where to look for information that could clarify their confusion. It would be helpful if the paper could provide a brief explanation in the main text on how this coupling is addressed, as it would improve the reader’s understanding.
    * The explanation of the training method of 'memory trigger' seems overly general and lacks sufficient detail. More details about how the RL training is conducted could be included in the Appendix.
2. More experiments to verify the necessity of 'memory waver' could be appreciated. Currently, the paper has conducted an ablation study about the value of 'memory trigger', but the corresponding ablation of 'memory waver' is absent.

**Questions:**

1. Could you please explain why you use hidden states **H** as the input of 'memory trigger' and 'memory waver' instead of the original text? It seems odd that the hidden states, which are already encoded by the LLM, go through another LLM one more time, especially since the LLM is only a LoRA variant version of the same LLM that **H** resulted from. The LLMs are learned to handle text tokens, and the space of encoded hidden states may differ substantially from the original space of token embedding.
2. I am curious about the memory capacity of your framework. Your sequential experiments for RQ3 have demonstrated that after training on the training data from the four datasets, your framework exhibits almost no 'forgetting', which is pretty promising. But will your framework eventually start to 'forget' when the training samples scale up? If so, what is the capacity of your framework? Moreover, how can we increase the capacity?

---

> ### Author Response · Authors · 2025-11-20
> **[Part 1/3] Author Response**
>
> We would like to express our deepest respect for your meticulous review! In response to your thoughtful comments, we have carefully prepared a point-by-point reply:
>
>
> ---
> > **`Weakness 1.1`** Sections 4.2 and 4.3 seemed confusing to me before reading Appendix C and D. Both 'memory trigger' and 'memory waver' require an instance of each other for their own training. This may confuse readers, and is only explained in Appendix D. Considering that Appendix D is not referenced in the main text until the end of Section 5.1, readers may find it difficult to identify where to look for information that could clarify their confusion. It would be helpful if the paper could provide a brief explanation in the main text on how this coupling is addressed, as it would improve the reader’s understanding.
>
>
> We sincerely apologize for any misunderstanding. To avoid further confusion, we have followed your suggestion and added a clear explanation of the weaver and trigger training procedure directly in the main text (Line 293), with an explicit reference to Appendix D. These changes have already been incorporated into the updated OpenReview manuscript PDF. Thank you again for your valuable feedback!
>
>
> ---
>
> > **`Weakness 1.2`** The explanation of the training method of 'memory trigger' seems overly general and lacks sufficient detail. More details about how the RL training is conducted could be included in the Appendix.
>
>
> Thank you once again for your valuable suggestion, which has significantly improved the completeness of our paper! In accordance with your recommendation, we have added the full RL training pipeline for the memory trigger in Appendix C.3 of the revised manuscript.
>
>
>
> ---
> > **`Weakness 2`** More experiments to verify the necessity of 'memory waver' could be appreciated. Currently, the paper has conducted an ablation study about the value of 'memory trigger', but the corresponding ablation of 'memory waver' is absent.
>
> Thank you very much for this insightful comment. In reponse, we further ablate the contribution of the memory weaver by introducing the *pause token* baseline (ICLR 2024) [1]. This method learns a small set of pause tokens that instruct the model to temporarily suspend outward generation and internally process the hidden state sequence, without producing visible output. A key distinction from MemGen is that our memory weaver generates context-dependent latent memory tokens, while pause tokens are fixed and context-agnostic.
>
> For this reason, we adopt the following baseline: whenever the trigger decides that memory should be activated, we replace the memory weaver with the insertion of $K$ learned pause tokens. We believe this baseline fairly captures the role of the weaver and highlights the importance of dynamically reconstructing memory from context. (if you feel there is a more suitable implementation/ablation strategy, we would be very happy to hear your suggestions!)
>
> The detailed training procedure for pause tokens has been added to Appendix E.3.4. We report the ablation results below:
>
> *Ablation study of the memory weaver on Qwen2.5-1.5B-Instruct.*
> | Method | Kodcode | TriviaQA | GPQA |
> |:-|:-|:-|:-|
> | Trigger + Pause Token | 49.50 | 56.30 | 13.80 |
> | Trigger + Weaver | 58.16 | 65.02 | 18.28 |
>
>
>
> We observe that although the trigger + pause token setting also yields performance improvement, it still lags significantly behind the trigger + weaver configuration. We believe this is because a small, fixed set of pause tokens cannot provide the same level of context-dependent support for reasoning as the memory weaver. A more detailed analysis is provided in Appendix E.3.4.

---

> ### Author Response · Authors · 2025-11-20
> **[Part 2/3] Author Response**
>
> > **`Question 1`**: Could you please explain why you use hidden states H as the input of 'memory trigger' and 'memory waver' instead of the original text? It seems odd that the hidden states, which are already encoded by the LLM, go through another LLM one more time, especially since the LLM is only a LoRA variant version of the same LLM that H resulted from. The LLMs are learned to handle text tokens, and the space of encoded hidden states may differ substantially from the original space of token embedding.
>
> Thank you for your professional insights! We would like to explain the motivation behind this design choice from two perspectives:
>
> * **Information loss:** We feed the trigger and weaver the full hidden-state matrix $\mathbf{H}_{t,<j}$ rather than decoding it back into token IDs because the latter would inevitably discard representational detail. Hidden states encode the model’s *unmanifest* internal computations (intermediate reasoning intent,  contextual nuances) that are not recoverable from explicit tokens. Consider a very toy example, the prompts *“Is 1+1=2?”* and *“Is 1+1=3?”* may elicit the same first output token (“Oh”), yet their hidden states, and the following outputs, differ substantially (e.g., *“Oh, that is surely right”* versus *“Oh, the user seems to be joking”*). Decoding $\mathbf{H}\_{t,<j}$ into discrete token IDs would collapse these differences, whereas operating directly in the hidden-state space preserves the rich contextual signals that the trigger and weaver are designed to leverage. To further clarify this design rationale, we have added an explicit explanation at Line 230 in the revised manuscript.
> * **Mechanism compatibility:** We acknowledge that directly using hidden states as input embeddings may create a mismatch with the reasoner’s token-embedding space. This issue has been recognized in prior latent-reasoning systems as well: for instance, SoftCoT introduces a linear layer to map hidden states into the token-embedding space, and Latent-SFT constrains latent embeddings to be linear combinations of existing token embeddings. Following these successful designs, our weaver applies a linear projection to map its LoRA-generated hidden states into the reasoner’s token-embedding space. This component corresponds to the `self.weaver_to_reasoner` module in our anonymous code repository (`larm/memory_generator/memgen_model.py`, Line 143). We have added an explicit explanation of this mechanism at Line 274 in the revised manuscript for better clarity.
>
> Thank you again for your critical insights and expertise!

---

> ### Author Response · Authors · 2025-11-20
> **[Part 3/3] Author Response**
>
> > **`Question 2`**: I am curious about the memory capacity of your framework. Your sequential experiments for RQ3 have demonstrated that after training on the training data from the four datasets, your framework exhibits almost no 'forgetting', which is pretty promising. But will your framework eventually start to 'forget' when the training samples scale up? If so, what is the capacity of your framework? Moreover, how can we increase the capacity?
>
> Thank you very much for this insightful comment. We agree that understanding the memory capacity of MemGen is important, and we clarify here that MemGen does have a finite capacity in a domain-adaptive manner (rather than as catastrophic forgetting).
>
> To examine this more concretely, we extend the sequential experiment from RQ3 (Table 4) by continuing training on the science subset of the `nvidia/Llama-Nemotron-Post-Training-Dataset`. Starting from the checkpoints reported in Table 4 (after AQuA/GPQA/GSM8K/KodCode), we continue training with successive batches of 2k science examples and track performance across tasks:
>
> *[Continual training on the science subset after AQuA/GPQA/GSM8K/KodCode; each row “1–7” denotes one additional 2k-step training round.]*
> | | AQuA | GPQA | GSM8K | KodCode |
> |---|---|---|---|---|
> | Vanilla | 41.34| 11.62| 39.51|24.55|
> | 1 | 42.01 | 29.80 | 50.85 | 48.85 |
> | 2 | 44.49 | 29.29 | 59.90 | 45.05 |
> | 3 | 50.00 | 30.30 | 62.85 | 43.85 |
> | 4 | 48.22 | 30.79 | 59.11 | 40.70 |
> | 5 | 47.90 | 32.85 | 60.51 | 31.10 |
> | 6 | 42.67 | 31.78 | 56.46 | 32.74 |
> | 7 | 41.10 | 32.29 | 47.22 | 30.65 |
>
>
> We observe that forgetting emerges gradually and is strongly domain-dependent:
> - GPQA remains stable or improves because its domain is aligned with the science subset.
> - AQuA and GSM8K (math reasoning) improve for approximately five rounds but begin to degrade in rounds 6–7 as the weaver increasingly specializes toward the science domain.
> - KodCode (coding), which is furthest from science-domain structure, begins to decline earliest, around round 5.
>
> Importantly, even at round 7, performance on most tasks remains above the vanilla baseline, indicating slow, controlled forgetting rather than catastrophic collapse. This validates that MemGen retains information more robustly than standard parametric finetuning, yet still ultimately has finite representational capacity.
>
> Regarding how to increase MemGen’s memory capacity, we view this as a highly promising direction for future work. Since the weaver is parameterized by a fixed-size LoRA module, infinite accumulation is not realistic. However, memory recruiting mechanisms, such as allocating additional LoRA blocks and composing MoE of weavers when existing capacity saturates, could allow MemGen to scale its latent memory pool dynamically. Once again, we are sincerely grateful for raising this thoughtful and important point.
>
> ---
>
>
> [1] Think before you speak: Training Language Models With Pause Tokens. ICLR 2024
> [2] Training Large Language Models to Reason in a Continuous Latent Space, COLM 2025
> [3] CODI: Compressing Chain-of-Thought into Continuous Space via Self-Distillation, CODI 2025
> [4] SoftCoT: Soft Chain-of-Thought for Efficient Reasoning with LLMs, ACL 2025
> [5] Latent Reasoning in LLMs as a Vocabulary-Space Superposition

---

### Official Review · Reviewer_KDeH · 2025-10-30

**Soundness:** 3
**Presentation:** 3
**Contribution:** 3
**Rating:** 4
**Confidence:** 4

**Summary:**

The paper proposes MemGen, a memory module for LLM agents: A small trigger watches the model’s hidden states and decides when to call memory. A weaver then generates a short sequence of latent vectors (soft tokens) that are inserted into the model’s hidden states to guide the next steps of reasoning. The base model stays frozen. The authors test MemGen on many benchmarks and report solid gains over parametric finetuning and retrieval memory. They also analyze the learned latent vectors and suggest roles like planning, procedural, and working memory.

**Strengths:**

-  **Originality:** Although the paper did not fully convince me about the usefulness of the mechanism, I found the work creative and inspiring. A big difference between LLMs and humans is that humans can invoke on-demand details based on a cue, effectively interrupting ongoing reasoning to retrieve distributed multimodal representations that support the current task (for example, recalling the visual details of a face when talking about a person). I have not seen this specific on-demand recall framing in prior work, and it could be more useful in other settings too (for example, human-like cross-modal recall).

**Weaknesses:**

* **Unfair comparison (most critical):** The paper compares against baselines with the same backbone size as the reasoner. However, MemGen uses two extra components that are instantiated on the backbone (trigger and weaver, implemented with LoRA). This increases effective capacity and compute. A fairer comparison would use three base models with LoRA adapters or a single larger model with a comparable parameter and compute budget. Models with larger capacities could implement the same mechanisms internally, reducing the need for an artificial extra trigger (the trigger could be "implicitly" learned inside the larger model).

* **Unclear mechanisms and training signals:** The paper says the synthesized memories are prepended to the internal state of the frozen reasoner, but it is not clear where this happens, which layer sees them, and how they interact with the existing hidden states and KV cache. The reward is also not explained, and it is unclear where the training signals and datasets come from.

* **Baseline setup:** The baselines (for example, SFT, GRPO, REINFORCE) feel under‑specified, and for the memory baselines it is uncertain they are the best comparison for these tasks. Do the chosen tasks actually require a memory system, or would stronger reasoning or longer context suffice?

**Questions:**

* **Reward and data:** How are rewards calculated for both the trigger and the weaver training? Where do the training signals come from? Is another AI judging the results, or are there gold labels or environment success signals? Please also clarify the sources of the training data.

* **Capacity vs necessity:** Is there anything MemGen can learn that a model cannot internalize directly in its parameters with the same trainable‑parameter and compute budget? Today’s models already decide when to stop thinking and how to reuse past context to some degree.

* **Nature of the latent outputs:** Do you have any explanation or intuition about the resulting non-human-readable latent tokens and why they are effective?

* **Exact insertion details:** When you say memories are prepended to the internal state, where exactly is this done, at which layer, and how are they combined with existing states and attention caches?

---

> ### Author Response · Authors · 2025-11-20
> **[Part 1/3] Author Response**
>
> We sincerely thank you for the thoughtful and constructive reviews of our manuscript! Based on your questions and recommendations, we give point-by-point responses to your comments and describe the revisions we made to address them.
>
> ---
> > **`Weakness 1`**: Unfair comparison (most critical): The paper compares against baselines with the same backbone size as the reasoner. However, MemGen uses two extra components that are instantiated on the backbone (trigger and weaver, implemented with LoRA). This increases effective capacity and compute. A fairer comparison would use three base models with LoRA adapters or a single larger model with a comparable parameter and compute budget.
>
>
> Thank you for this thoughtful question! We would like to respond to your concern from three perspectives (and we apologize in advance if we misunderstood any part of your intended meaning; please feel free to correct us):
>
> * **MemGen’s trainable parameter count is *far smaller* than most baselines.**
>   Recall that our baselines include SFT, GRPO, and REINFORCE, all of which involve full-parameter tuning. Assuming an 8B-parameter reasoner backbone, the number of trainable parameters is **8B**. In contrast, MemGen introduces two LoRA adapters with rank 16 attached to the reasoner; for an 8B model, MemGen has only **16.8M** trainable parameters, i.e., **just 0.21%** of the trainable parameters used by SFT, GRPO baselines.
> * **MemGen does introduce extra learnable parameters, but this practice seems to be widely accepted in the community.**
>   A large body of LoRA-related work, e.g., LoRA (ICLR 2022), LoRA+ (ICML 2024), AdaLoRA (ICLR 2023), directly compares (1) fine-tuning with a LoRA adapter versus (2) full-parameter SFT,
>   without treating the introduction of LoRA parameters as an unfair comparison.
>   More recent works such as LoRAMoE (ACL 2024) and MoLA (ACL 2025) use 5+ LoRA adapters, yet still compare against SFT baselines that introduce *zero* extra parameters. Thus, we humbly believe that the prevailing community standard treats “adding a small number of extra parameters while keeping trainable parameters far below the baseline” as a fair evaluation paradigm.
> * **In line with your wise suggestion, we have added LoRA-based baselines.** To honor your thoughtful recommendation, we additionally include LoRA+SFT and LoRA+GRPO baselines. Since MemGen employs two LoRA modules with rank 16 and a total of 16.8M additional parameters, we set the LoRA rank for LoRA+SFT and LoRA+GRPO to 64, which yields 16.5M extra parameters. This ensures that their extra learnable parameter budgets are closely comparable.
>
> *Performance comparison conducted on Qwen2.5-1.5B.*
> | Method  |Extra parameters  | Kodcode | TriviaQA |
> |--|--|-|-|
> | SFT+LoRA (r=64)  | 16.5M | 55.51  | 63.20  |
> | GRPO+LoRA (r=64) |16.5M  | 61.10  | 66.00   |
> | MemGen-S (LoRA r=16) | 16.8M | 58.16   | 65.02   |
> | MemGen-G (LoRA r=16) |16.8M | 65.43   | 73.42  |
>
> Even under a comparable extra learnable parameter budget, MemGen consistently outperforms the baseline methods. We believe this clearly indicates that MemGen’s advantage does not stem merely from additional parameters, but from its unique latent memory mechanism.
>
>
> ---
>
> > **`Weakness 2.1: Unclear mechanisms`** Unclear mechanisms and training signals: The paper says the synthesized memories are prepended to the internal state of the frozen reasoner, but it is not clear where this happens, which layer sees them, and how they interact with the existing hidden states and KV cache.
>
> Thank you for the excellent question! Please allow us to address each point in turn:
>
>
> * **Where are the latent memory tokens inserted?**
>   We append the generated latent memory tokens to the prefix’s input-embedding sequence. That is, the latent memory is treated as additional (virtual) tokens at the embedding level.
> * **Which layer sees them?**
>   Because insertion happens at the input-embedding layer, the latent memory tokens are visible to all subsequent transformer layers of the frozen reasoner.
> * **How do they interact with the existing hidden states and KV cache?**
>   Let $E \in \mathbb{R}^{T \times d}$ denote the frozen reasoner’s **input-embedding sequence** for the current prefix. At a trigger step, the weaver generates $M \in \mathbb{R}^{K \times d}$ as the latent-memory matrix. We form the extended embedding sequence $\hat{E} = [\,E; M\,] \in \mathbb{R}^{(T+K) \times d}$ and extend the attention mask accordingly. The next decoding step of the frozen reasoner is executed with inputs_embeds $= \hat{E}$ and attention_mask $= \hat{A}$. Under a standard transformer, each layer $\ell$ constructs $K\_\ell = W^K\_\ell \hat{H}\_\ell, V\_\ell = W^V\_\ell \hat{H}\_\ell$, where $\hat{H}\_\ell$ is the layer-$\ell$ hidden-state sequence produced from $\hat{E}$. Thus, the $K$ latent memory vectors simply contribute $K$ additional rows to the KV cache and are attended to exactly like ordinary tokens, requiring no special KV handling.

---

> > ### Comment · Reviewer_KDeH · 2025-11-26
> >
> > Thank you for your answers and your clarification on insertion, and sorry for not having been clearer. My concern was about inference not training, the latter is obvious. LoRA adapters are useless without the base model, so in the most naive scenario, you'd have to use 3 times the base model + LoRA adapter depending on trigger/weaver etc.
> > So my question was : at inference time, e.g; for a single generation step where memory is invoked, how many forward passes through transformer layers occur?
> > Specifically:
> > Does the trigger require a forward pass through the entire base model (with trigger LoRA)?
> > Does the weaver require a forward pass through the base model (with weaver LoRA)?
> > Does the reasoner then do its own forward pass (frozen)?
> > Or do they somehow happen sequentially within the same pass?
> > If yes to all three, then MemGen uses 3x the inference compute of baselines, regardless of parameter count. If everything happens in a single forward pass, could you explain or remind how?
> >
> > If you somewhat use three models, then the more reasonable baseline I had in mind is for example ensembling 2 models (with LoRA if you wish), the first adapter is a planner, and the second executes the strategy, then run twice, one to plan, the second to execute the planning.

---

> > > ### Author Response · Authors · 2025-11-27
> > >
> > > Thank you for your engagement and kind clarification! We address the concern from the following three perspectives:
> > >
> > > ---
> > > **1. Fundamentally, MemGen never approaches $3\times$ the vanilla model inference budget.**
> > >
> > > We now better understand the reviewer’s concern: if both the trigger and the weaver ran at every decoding step, the total cost might appear to reach $3 \cdot c_\text{base}$ ($c_\text{base}$ is the vanilla inference cost). We clarify that this never occurs, because, as stated on Line 241, the trigger follows a sentence-granularity activation strategy: it runs only on a delimiter token (comma, period, newline, etc.). Thus, each token falls into one of three cases:
> > >
> > >  - **One forward pass (vast majority)**: the token is not a delimiter. no extra computation,
> > >  - **Two forward passes (minority)**: the token is a delimiter, but the trigger chooses not to invoke memory.
> > >  - **Three forward passes (minority)**: the token is a delimiter and the trigger invokes the weaver.
> > >
> > > Because most tokens in natural-language reasoning are non-delimiters, the first case overwhelmingly dominates.
> > >
> > > So, how much extra inference cost does MemGen incur in real scenarios? Formally, let a trajectory contain $T$ tokens and normalize $c_\text{base}=T$. Let $b_j\in\{0,1\}$ mark whether the $j$-th token is a delimiter, and $d_j\in\{0,1\}$ mark whether memory is invoked there. Counting one unit for each trigger/weaver forward, the total cost is
> > >
> > > $$
> > > c_\text{memgen}=\sum_{j=1}^T (1+b_j+b_j d_j).
> > > $$
> > >
> > > Let
> > >
> > > $$
> > > D=\sum_{j=1}^T b_j,\qquad I=\sum_{j=1}^T b_j d_j,
> > > $$
> > >
> > > and define the delimiter ratio $\rho=D/T$ and invocation ratio $\alpha = I/D$ (with $\alpha=0$ if $D=0$). Then
> > > $$
> > > \frac{c_\text{memgen}}{c_\text{base}}
> > > = 1 + \rho + \alpha\rho
> > > = 1 + (1+\alpha)\rho.
> > > $$
> > >
> > > Since $\alpha\le 1$,
> > >
> > > $$
> > > \frac{c_\text{memgen}}{c_\text{base}} \le 1 + 2\rho \ll 3,
> > > $$
> > >
> > > because natural-language outputs satisfy $\rho \ll 1$. Hence, even under the conservative assumption that each trigger/weaver pass costs as much as a full base forward, MemGen is *structurally* bounded far below the pathological $3\times$ scenario.
> > >
> > > **Empirically**, the bound is even tighter. On GSM8K, we observe $\rho \approx 0.08$ and $\alpha \approx 0.6$, yielding
> > >
> > > $$
> > > \frac{c_\text{memgen}}{c_\text{base}}
> > > = 1 + (1+0.6)\times 0.08
> > > \approx 1.128.
> > > $$
> > >
> > > Thus, MemGen operates at only **1.13×** the vanilla budget, dramatically closer to $1$ than to $3$.
> > >
> > >
> > > ---
> > >
> > > **2. In practice, MemGen’s inference budget remains far below $3\times$ the base model.**
> > >
> > > Beyond the theoretical bound, we also observe clear empirical evidence that MemGen’s inference-time overhead is modest. As shown in Table 8, MemGen’s per-query latency is nearly identical to the SFT baseline: on TriviaQA+SmolLM-3B, SFT and MemGen take **3.05 s** and **3.16 s**, respectively. Notably, even the *vanilla* base model is slower (**4.26 s**), confirming that MemGen’s overhead is far below the hypothetical $3\times$ multiplier. Empirically, MemGen behaves exactly as predicted theoretically: its inference cost remains tightly concentrated near **$1\times$**, not **$3\times$**.
> > >
> > >
> > > ---
> > >
> > > **3. We respectfully include the reviewer’s suggested baseline.**
> > >
> > > Following the reviewer’s suggestion, we additionally compare MemGen against baselines that *explicitly* incur significantly higher inference budgets:
> > >
> > > * **ReMA (NeurIPS 2025) [1]**: This method closely matches the reviewer’s proposed design—two full-parameter RL-finetuned models, one for high-level planning and one for execution. Inference requires running the planner once and the executor once, yielding an inference cost of ≈2× the base model.
> > > * **Majority voting**: We ensemble $k$ independently LoRA+SFT finetuned models and sample $M$ responses per model, aggregating via majority vote. The inference budget is therefore $k \cdot M$ times the base model’s cost, which rapidly exceeds MemGen’s budget.
> > >
> > > *[Peformance comparison using identical training data (MATH 7.5k) and a shared backbone (Qwen2.5-7B)]*
> > > |Method| Trainable Params.|Inference cost vs. base|Per-Query FLOPs|MATH|GSM8K|Minerva Math |
> > > |-|-|-|-|-|-|-|
> > > | ReMA|2*7B=14B|≈2×|453.1 TFLOPs     | 74.4 | 90.6| 34.9|
> > > | Voting (k=1, M=3) | 8.1M  | ≈3× | 668.0 TFLOPs| 72.2     | 84.3     | 29.7|
> > > | Voting (k=3, M=3) | 24.3M | ≈9× | 2036.8 TFLOPs    | 73.6| 87.4     | 30.5         |
> > > | MemGen-SFT| 14.2M| ≈1.2×                | 217.6 TFLOPs | 76.8 | 90.9 | 34.2     |
> > >
> > > MemGen requires *far fewer* trainable parameters and *substantially lower* per-query FLOPs than both ReMA and majority-voting (3,3), while achieving comparable or superior accuracy across all benchmarks. We believe this constitutes a fair and rigorous comparison, demonstrating that **MemGen provides strong gains with far less inference budget than the baselines**.
> > >
> > >
> > > ----
> > >
> > > [1] ReMA: Learning to Meta-think for LLMs with Multi-agent Reinforcement Learning

---

> ### Author Response · Authors · 2025-11-20
> **[Part 2/3] Author Response**
>
> > **`Weakness 2.3 & Question 1.2: training datasource`** it is unclear where the training signals and datasets come from; Please also clarify the sources of the training data.
>
> Thank you for the question! As elaborated in Appendix D.1, for all datasets except PopQA, we use their training splits to train both the memory weaver and the memory trigger. For other baselines that require training (SFT, GRPO, REINFORCE, SoftCoT, Co-processor, etc.), we adopt **exactly the same data splits**. For baselines that do not require training (e.g., ExpeL), we construct the memory database using the training split and evaluate on the test split. To make this clearer, we have added the following data summary table:
>
>
> | Dataset Name |Train Size | Test Size | Note|
> |-|-|-|-|
> | ALFWorld | 3.32K | 134 | From AgentBank-ALFWorld	|
> | TriviaQA | 4.13K | 7.9K |From AgentBank-TriviaQA|
> | KodCode | 8K | 2K | KodCode-Light-RL-10K |
> | BigCodeBench | 912 | 228 | -|
> | GPQA | 448 | 198 | GPQA-Diamond for testing|
> | GSM8K | 7.47K | 1K |- |
> | MATH |1.6K | 500| -|
>
> We have also updated the manuscript to include a dataset statistics table in Appendix D.1.
>
> ---
>
> > **`Weakness 3: Baseline setup`** Baseline setup: The baselines (for example, SFT, GRPO, REINFORCE) feel under‑specified, and for the memory baselines it is uncertain they are the best comparison for these tasks. Do the chosen tasks actually require a memory system, or would stronger reasoning or longer context suffice?
>
> To address your concern, we justify our choice of baselines point by point:
>
> * **SFT, GRPO, and REINFORCE** serve as the most direct comparisons: each baseline performs full-parameter fine-tuning on the training split of every dataset. Given that all methods use the *same* training data, while these baselines have roughly 470× more trainable parameters than MemGen, we believe they constitute fair and meaningful baselines.
> * **Retrieval-based memory systems** such as ExpeL (AAAI 2024 Oral), MemoryBank (AAAI 2024), and AWM (ICML 2025) are well-established memory baselines. In fact, ExpeL paper directly adopts ALFWorld (which we also use) and on QA tasks closely related to ours, such as HotpotQA, which is directly comparable to PopQA and TriviaQA.
>
> We humbly believe that our chosen tasks are well-grounded in prior studies:
>
> * **ALFWorld** has been widely used to evaluate agent-memory capabilities, including in ExpeL (AAAI 2024), CDMem (NAACL 2025) [1], and AgentSquare (ICLR 2025) [2].
> * **TriviaQA and PopQA** belong to the family of knowledge-intensive QA tasks; works such as G-Memory (NeurIPS 2025) [3] and [6] also use these QA datasets to evaluate agent memory.
> * **KodCode and BigCodeBench** represent coding benchmarks frequently adopted in self-evolving agent and agent-memory papers, such as EvoMAC (ICLR 2025) [4].
> * **MATH and GPQA**, which emphasize reasoning capacities, are similarly used by memory papers such as Dynamic Cheatsheet [5].
>
> In summary, we believe that the selected baselines and benchmarks are appropriate and well supported by prior literature. We respectfully hope this helps address your concern.
>
>
> ---
> > **`Question 2: Capacity vs necessity`**  Is there anything MemGen can learn that a model cannot internalize directly in its parameters with the same trainable‑parameter and compute budget? Today’s models already decide when to stop thinking and how to reuse past context to some degree.
>
> We appreciate the reviewer’s question regarding why MemGen introduces two additional modules instead of internalizing this ability into a larger model. We clarify our design rationale as follows:
>
> **(1) Lightweight and modular design choice.**
> MemGen introduces the trigger–weaver modules because they offer a practical and lightweight way to endow off-the-shelf LLMs with dynamic memory capabilities without modifying backbone parameters. This modularity is itself valuable: it avoids catastrophic forgetting, ensures stable optimization, and allows any off-the-shelf model to immediately benefit from generative memory.
>
>
> **(2) Our core contribution lies in latent, generative memory.**
> Another core contribution of MemGen lies in formulating memory as latent tokens that are dynamically synthesized, compressed, and inserted into reasoning. As the you wisely notes, “today’s models already decide when to stop thinking and how to reuse past context to some degree.” We believe such behaviors correspond to systems like Mem1 and MemAgent, which indeed make progress in adaptive memory control but still rely on explicit, human-readable text memories, rather than reconstructive, machine-native latent memory. To the best of our knowledge, no prior work provides a generative latent memory mechanism of this form. The external trigger-weaver in MemGen serves as one effective realization of this idea, but do not define the idea itself.

---

> ### Author Response · Authors · 2025-11-20
> **[Part 3/3] Author Response**
>
> **(3) We do believe internalization is feasible and a promising future direction.**
> We fully agree that larger models could internalize these abilities, for example, implementing trigger via a special `<memory_trigger>` token. Exploring when and how a backbone can autonomously synthesize latent memory tokens is an exciting future direction (though also comes with challenges such as preventing catastrophic forgetting and balancing latent reasoning with explicit reasoning). Our modular design is simply one effective instantiation of the *generative latent memory* paradigm we advocate, and we sincerely hope future work will explore additional ways to realize this broader idea.
>
>
> ---
> > **`Question 3: Nature of the latent outputs`** Do you have any explanation or intuition about the resulting non-human-readable latent tokens and why they are effective?
>
> We would like to humbly point out that, the entirety of **Section 5.3** is devoted to providing exactly the type of *explanation and intuition* you requested regarding the “non-human-readable latent tokens.” Please allow us to summarize our analysis more clearly:
>
> * In **Appendix G**, we present a large corpus of hard-decoded latent memory tokens and show that they are indeed **human-unreadable**.
> * On **Line 418 (Section 5.3), Figures 5 & 11**, we cluster latent memory tokens across datasets and within the same dataset and observe **clear, meaningful clustering structures**: tokens from the same cluster often share structural patterns (e.g., many clusters contain tokens that end with “_check”), suggesting the presence of consistent latent functions.
> * In **Line 430 and Figure 6 (Right)**, to further interpret *why* these clusters exist and *what functional roles* they play, we conduct a **post-hoc intervention study**. The results show that different clusters correspond to different memory functionalities, including planning memory, procedural memory (e.g., guidance for answer formatting), working memory(e.g., short-term handling of intermediate reasoning steps)
>
> These findings suggest that although the latent tokens are not directly interpretable, they nonetheless encode **distinct and meaningful cognitive roles**.
>
> We are also grateful that this analysis was acknowledged by Reviewer *KDeH* and Reviewer *Z9t6*. If you have any further or more specific questions, we would be more than honored to elaborate.
>
> ---
>
> > **`Question 4: Exact insertion details`** When you say memories are prepended to the internal state, where exactly is this done, at which layer, and how are they combined with existing states and attention caches?
>
> Thank you for the insightful question! Please refer to our response to `Weakness 2.1`.
>
>
>
> ---
>
> [1] An Efficient Context-Dependent Memory Framework for LLM-Centric Agents, NAACL 2025
> [2] AgentSquare: Automatic LLM Agent Search in Modular Design Space, ICLR 2025
> [3] G-Memory: Tracing Hierarchical Memory for Multi-Agent Systems, NeurIPS 2025
> [4] Self-Evolving Multi-Agent Collaboration Networks for Software Development, ICLR 2025
> [5] Dynamic Cheatsheet: Test-Time Learning with Adaptive Memory
> [6] From Experience to Strategy: Empowering LLM Agents with Trainable Graph Memory

---

### Official Review · Reviewer_hdQW · 2025-10-31

**Soundness:** 3
**Presentation:** 3
**Contribution:** 3
**Rating:** 8
**Confidence:** 5

**Summary:**

The paper presents MemGen, a generative latent memory framework that enables LLM agents to dynamically synthesize and integrate memory during reasoning. The core contribution—a memory trigger coupled with a memory weaver—is well-motivated and technically sound, offering a promising alternative to parametric and retrieval-based paradigms. Experiments across nine benchmarks demonstrate consistent gains and strong cross-domain generalization, with ablations supporting the necessity of learned memory invocation.

**Strengths:**

* I find it novel to leverage latent tokens as memory carriers, given that only two LoRA adapters are fine-tuned while surpassing full GRPO. The proposed method differs from previous latent-based memory approaches such as MemoryLLM and Memory3 in that the latent memory is context-sensitive rather than reusing KV caches. This aligns with the authors’ concept of generative memory.
* The method demonstrates promising transferability (Figure 3); inserting tokens to intervene in decoding appears to effectively mitigate catastrophic forgetting. The frequency study in Figure 4 is interesting.
* The authors manage to provide interpretability analyses of latent memory, ranging from clustering visualizations to post-hoc examinations.

**Weaknesses:**

* I am somewhat confused about the memory form: it seems to reside within the weaver’s parametric memory, suggesting that memgen may not be a purely latent-memory method—its carrier is parametric knowledge manifested in latent form.
* How are the weaver and trigger trained? Is the process akin to an EM-style alternating optimization, or are they jointly trained? If the latter, the training process risks to be highly unstable.
* More ablation studies could be included, particularly regarding the LoRA parameter settings of the weaver (Table 6 shows only a single configuration).
* Could the latent token length be made dynamic rather than fixed a priori?

**Questions:**

- How is the trigger and weaver jointly trained?
- Does fixing K impose a capacity bottleneck on more long-horizon task? Is it possible that memgen can benefit from a dynamic memory length mechanism?
- How is catastrophic forgetting avoided inside the weaver itself, since its parameters continuously absorb new experience while the core LLM stays frozen?
- Could the authors provide more detailed ablation studies?

---

> ### Author Response · Authors · 2025-11-20
> **[Part 1/2] Author Response**
>
> Thank you immensely for your time and efforts, as well as the helpful and constructive feedback! Here, we give point-by-point responses to your comments.
>
> ---
>
> > **`Weakness 1`**: I am somewhat confused about the memory form: it seems to reside within the weaver’s parametric memory, suggesting that memgen may not be a purely latent-memory method—its carrier is parametric knowledge manifested in latent form.
>
> Thank you for your insightful observation. We agree that, indeed, the knowledge encoded in a Weaver model can be viewed as a form of parametric memory. However, we would like to kindly recall that MemGen also supports external retrieval-based memory as an alternative carrier (as discussed in Line 296). Regardless of whether the underlying knowledge originates from parametric storage or retrieval-based sources, both ultimately operate through a unified interface, namely the latent memory tokens, which are generated and consumed during generation. Therefore, we respectfully view MemGen as fundamentally a latent-memory framework, while noting that it conveniently bridges multiple underlying memory modalities.
>
> ---
>
> > **`Weakness 2 & Question 1`**: How are the weaver and trigger trained? Is the process akin to an EM-style alternating optimization, or are they jointly trained? If the latter, the training process risks to be highly unstable.
>
> Thank you for your question. As we clarified in Appendix D.1, the weaver and trigger components of MemGen are trained separately. Concretely, we first train the memory weaver. Since the trigger does not exist at this stage, we substitute it with a simple random decision function (which inserts memory at random semantic boundaries) as a proxy for trigger. After the weaver is fully trained, we freeze it and then proceed to train the trigger. We believe this simpler procedure helps stabilize training, and in practice it is more robust than an EM-style alternating optimization or a fully joint-training scheme, as you suggested.
>
> Thank you again for raising this point. To make this clearer and avoid misunderstanding, we have strengthened the explanation in Line 293 of the updated manuscript.
>
> ---
>
> > **`Weakness 3`**: More ablation studies could be included, particularly regarding the LoRA parameter settings of the weaver (Table 6 shows only a single configuration).
>
>
>
> Thank you for this insightful query! To examine the sensitivity of MemGen’s weaver to the LoRA parameter setting, we additionally conducted experiments with
> $r \in \\{4, 6, 8, 16, 24, 32\\}$, as shown below:
>
> *[LoRA sensitivity analysis]. The backbone LLM is selected as Qwen2.5-1.5B-Instruct.*
> | Method | GPQA   | Kodcode | TriviaQA |
> |-|-|-|-|
> | Weaver LoRA(r=4)| 17.16% | 54.85%  | 63.04%   |
> | Weaver LoRA(r=6)  | 18.18% | 55.25%  | 64.55%   |
> | Weaver LoRA(r=8)  | 17.67% | 55.75%  | 64.10%   |
> | Weaver LoRA(r=16) | 18.28% | 56.16%  | 65.02%   |
> | Weaver LoRA(r=24) | 18.67% | 57.20%  | 65.40%   |
> | Weaver LoRA(r=32)  | 19.26% | 57.95%  | 65.85%   |
> | Full SFT on Weaver | 21.21% | 60%     | 67.10%   |
>
> We observe that increasing the LoRA rank of the weaver naturally expands its parameter capacity and correspondingly yields additional performance gains for MemGen. However, the default configuration used in our experiments, LoRA with r=16, already achieves strong performance. We therefore adopt this setting consistently throughout the paper.
>
> ---
>
> > **`Weakness 4 & Question 2`**: Could the latent token length be made dynamic rather than fixed a priori? ... Does fixing K impose a capacity bottleneck on more long-horizon task? Is it possible that memgen can benefit from a dynamic memory length mechanism?
>
>
> Thank you for this excellent idea! In fact, we believe it is entirely feasible. There already exist well-developed techniques in the latent-reasoning literature that enable dynamic latent-token sequence lengths. For example, the latent end token mechanism introduced in [1] adds a trainable tensor in the hidden-state space; when the final hidden state approaches this tensor, the decoder automatically resumes text-token generation.
>
> For simplicity and clarity of training, we adopted a fixed-length implementation in our work. However, these techniques can be incorporated by readers with minimal engineering overhead. We sincerely hope this addresses your concerns.

---

> ### Author Response · Authors · 2025-11-20
> **[Part 2/2] Author Response**
>
> > **`Question 3`**: How is catastrophic forgetting avoided inside the weaver itself, since its parameters continuously absorb new experience while the core LLM stays frozen?
>
>
> Thank you very much for this insightful comment! We would like to humbly argue that MemGen is not immune to capacity limits or forgetting; rather, our claim is that it tends to forget more slowly than purely parametric approaches such as full-parameter SFT.
>
> * **Empirical evidence of slower forgetting.**
>   As shown in Table 5 of the manuscript, when exposed to a sequence of tasks, MemGen preserves prior knowledge substantially better than SFT. For example, after training on three additional datasets following AQuA, MemGen’s performance on AQuA drops by only about 3%, whereas SFT suffers a 14% decrease. We attribute this to two key design choices: MemGen’s context-sensitive latent memory synthesis and the trigger’s domain-aware activation.
> * **MemGen still has finite memory capacity.**
>   To more directly examine capacity and forgetting, we perform a continual learning experiment using the science subset of the `nvidia/Llama-Nemotron-Post-Training-Dataset`. Starting from the checkpoints reported in Table 4 (after AQuA/GPQA/GSM8K/KodCode), we continue training with successive batches of 2k science examples and track performance across tasks:
>
> *[Continual training on the science subset after AQuA/GPQA/GSM8K/KodCode; each row “1–7” denotes one additional 2k-step training round.]*
> | | AQuA | GPQA | GSM8K | KodCode |
> |---|---|---|---|---|
> | Vanilla | 41.34| 11.62| 39.51|24.55|
> | 1 | 42.01 | 29.80 | 50.85 | 48.85 |
> | 2 | 44.49 | 29.29 | 59.90 | 45.05 |
> | 3 | 50.00 | 30.30 | 62.85 | 43.85 |
> | 4 | 48.22 | 30.79 | 59.11 | 40.70 |
> | 5 | 47.90 | 32.85 | 60.51 | 31.10 |
> | 6 | 42.67 | 31.78 | 56.46 | 32.74 |
> | 7 | 41.10 | 32.29 | 47.22 | 30.65 |
>
>
>   We observe that forgetting emerges in a domain-dependent manner. The science subset is closely aligned with GPQA, so GPQA performance continues to improve steadily. AQuA and GSM8K (math reasoning) initially benefit from the additional training for roughly five rounds, but begin to exhibit noticeable forgetting in rounds 6–7, likely because the weaver’s knowledge becomes overly biased toward the science domain. KodCode, a coding benchmark with markedly different format and domain characteristics, degrades earliest, at around round 5. Notably, even at round 7, most tasks still remain above the vanilla baseline, indicating that forgetting is gradual rather than catastrophic.
>
> We regard understanding and extending MemGen’s memory capacity as a very promising direction for future work. Given that LoRA-based weavers have a bounded number of trainable parameters, it is unrealistic to expect unbounded knowledge accumulation. However, mechanisms such as “memory recruiting,” for example via LoRA MoE, could expand MemGen’s effective capacity. We are grateful for your insightful suggestion and plan to consider this explicitly in our future work.
>
> ---
> > **`Question 4`**: Could the authors provide more detailed ablation studies?
>
> Thank you for the inquiry. Please refer to our response to `Weakness 3`, where we present a more detailed sensitivity on LoRA parameters.
>
>
> ----
>
> [1] LATENT VISUAL REASONING

---

### Meta-Review · Area_Chair_r64i · 2026-01-06

**Summary:**

This work studies LLM-based agents equipped with a newly proposed memory mechanism that modifies the latent state in a memory-dependent manner. The method is deliberately simple and memory-efficient, relying on a LoRA-based adaptation to avoid full parameter updates of the backbone model. The reviewers’ main concern centers on the fairness of the comparison between the proposed approach and a vanilla baseline with a larger model size. I believe this concern has been adequately addressed in the rebuttal through controlled and size-aware comparisons.

**Reviewer Concerns:**

**Reviewer hdQW:** all concerns have been addressed.

**Reviewer KDeH:** all concerns have been addressed.

**Reviewer iSzn:** all concerns have been addressed.

**Reviewer Z9t6:** all concerns have been addressed.

**Reviewer Scores:**

**Reviewer hdQW:** 8 → 8

**Reviewer KDeH:** 4 → 6

**Reviewer iSzn:** 8 → 8

**Reviewer Z9t6:** 8 → 8

---

### Decision · Program_Chairs · 2026-01-26

Accept (Poster)